# DISCOVERING LATENT CONCEPTS LEARNED IN BERT

**Fahim Dalvi**[◇*] **Abdul Rafae Khan**[†*] **Firoj Alam**[◇] **Nadir Durrani**[◇] **Jia Xu**[†] **Hassan Sajjad**[◇]
{faimaduddin,fialam,ndurrani,hsajjad}@hbku.edu.qa
◇Qatar Computing Research Institute, HBKU Research Complex, Doha 5825, Qatar

†School of Engineering and Science, Steven Institute of Technology, Hoboken, NJ 07030, USA
{akhan4,jxu70}@stevens.edu

## ABSTRACT

A large number of studies that analyze deep neural network models and their ability to encode various linguistic and non-linguistic concepts provide an interpretation of the inner mechanics of these models. The scope of the analyses is limited to pre-defined concepts that reinforce the traditional linguistic knowledge and do not reflect on how novel concepts are learned by the model. We address this limitation by discovering and analyzing latent concepts learned in neural network models in an unsupervised fashion and provide interpretations from the model's perspective. In this work, we study: i) what latent concepts exist in the pre-trained BERT model, ii) how the discovered latent concepts align or diverge from classical linguistic hierarchy and iii) how the latent concepts evolve across layers. Our findings show: i) a model learns novel concepts (e.g. animal categories and demographic groups), which do not strictly adhere to any pre-defined categorization (e.g. POS, semantic tags), ii) several latent concepts are based on multiple properties which may include semantics, syntax, and morphology, iii) the lower layers in the model dominate in learning shallow lexical concepts while the higher layers learn semantic relations and iv) the discovered latent concepts highlight potential biases learned in the model. We also release[1] a novel BERT ConceptNet dataset (BCN) consisting of 174 concept labels and 1M annotated instances.

## 1 INTRODUCTION

Interpreting deep neural networks (DNNs) is essential for understanding their inner workings and successful deployment to real-world scenarios, especially for applications where fairness, trust, accountability, reliability, and ethical decision-making are critical metrics. A large number of studies have been devoted towards interpreting DNNs. A major line of research work has focused on DNNs in interpreting deep Natural Language Processing (NLP) models and their ability to learn various pre-defined linguistic concepts (Tenney et al., 2019b; Liu et al., 2019a). More specifically, they rely on pre-defined linguistic concepts such as: parts-of-speech tags and semantic tags, and probe whether the specific linguistic knowledge is learned in various parts of the network. For example, Belinkov et al. (2020) found that lower layers in the Neural Machine Translation (NMT) model capture word morphology and higher layers learn long-range dependencies.

A pitfall to this line of research is its study of pre-defined concepts and the ignoring of any latent concepts within these models, resulting in a narrow view of what the model knows. Another weakness of using user-defined concepts is the involvement of human bias in the selection of a concept which may result in a misleading interpretation. In our work, we sidestep this issue by approaching interpretability from a model's perspective, specifically focusing of the discovering of latent concepts in pre-trained models.

Mikolov et al. (2013); Reif et al. (2019) showed that words group together in the high dimensional space based on syntactic and semantic relationships. We hypothesize that these groupings represent *latent concepts*, the information that a model learns about the language. In our approach, we cluster

---

*Equal contribution
[1]Code and dataset: https://neurox.qcri.org/projects/bert-concept-net.html

contextualized representations in high dimensional space and study the relations behind each group by using hierarchical clustering to identify groups of related words. We manually annotate the groups and assemble a concept dataset. This is the first concept dataset that enables model-centric interpretation and will serve as a benchmark for analyzing these types of models. While we base our study on BERT (Devlin et al., 2019), our method can be generically applied to other models.

We study: i) what latent concepts exist in BERT, ii) how the discovered latent concepts align with pre-defined concepts, and iii) how the concepts evolve across layers. Our analysis reveals interesting findings such as: i) the model learns novel concepts like the hierarchy in animal kingdom, demographic hierarchy etc. which do not strictly adhere to any pre-defined concepts, ii) the lower layers focus on shallow lexical concepts while the higher layers learn semantic relations, iii) the concepts learned at higher layers are more aligned with the linguistically motivated concepts compared to the lower layers, and iv) the discovered latent concepts highlight potential biases learned in the model.

There are two main contributions of our work: i) We interpret BERT by analyzing latent concepts learned in the network, and ii) we provide a first multi-facet hierarchical BERT ConceptNet dataset (BCN) that addresses a major limitation of a prominent class of interpretation studies. Our proposed concept pool dataset not only facilitates interpretation from the perspective of the model, but also serves as a common benchmark for future research. Moreover, our findings enrich existing linguistic and pre-defined concepts and has the potential to serve as a new classification dataset for NLP in general. BCN consists of 174 fine-grained concept labels with a total of 1M annotated instances.

## 2 WHAT IS A CONCEPT?

A concept represents a notion and can be viewed as a coherent fragment of knowledge. Stock (2010) defines concept as "a class containing certain objects as elements, where the objects have certain properties". Deveaud et al. (2014) considers latent concepts as words that convey the most information or that are the most relevant to the initial query. A concept in language can be anything ranging from a shallow lexical property such as character bigram, to a complex syntactic phenomenon such as an adjectival phrase or a semantic property such as an event. Here, we loosely define concept as ***a group of words that are meaningful*** i.e. can be clustered based on a linguistic relation such as lexical, semantic, syntactic, morphological etc. For example, names of ice-hockey teams form a semantic cluster, words that occur in a certain position of a sentence represent a syntactic concept, words that are first person singular verbs form a morphological cluster, words that begin with "anti" form a lexical cluster. We consider clusters of word contextual representations in BERT as concept candidates and the human-annotated candidates with linguistic meanings as our discovered latent concepts. However, if a human is unable to identify a relation behind a group of words, we consider it as *uninterpretable*. It is plausible that BERT learns a relation between words that humans are unable to comprehend.

## 3 METHODOLOGY

Consider a Neural Network (NN) model $\mathbb{M}$ with $L$ layers $\{l_1, l_2, ...l_l, ..., l_L\}$, where each layer contains $H$ hidden nodes. An input sentence consisting of $M$ words $w_1, w_2, ...w_i, ..., w_M$ is fed into NN. For the $i$-th word input, we compute the node output (after applying the activation functions) $y_h^l(i)$ of every hidden node $h \in \{1, ..., H\}$ in each layer $l$, where $\overrightarrow{y}^l(i)$ is the vector representation of all hidden node outputs in layer $l$ for $w_i$. Our goal is to cluster $\overrightarrow{y}^l$, the contextual representation, among all $i$-th input words. We call this clustering as latent concepts that map words into meaningful groups. We introduce an annotation schema and manually assign labels to each cluster following our definition of a concept. In the following, we discuss each step in detail.

### 3.1 CLUSTERING

Algorithm 1 (Appendix A) presents the clustering procedure. For each layer, we cluster the hidden nodes into $K$ groups using agglomerative hierarchical clustering (Gowda & Krishna, 1978) trained on the contextualized representation. We apply Ward's minimum variance criterion that minimizes the total within-cluster variance. The distance between two vector representations is calculated with the squared Euclidean distance. The number of clusters $K$ is a hyperparameter. We empirically set

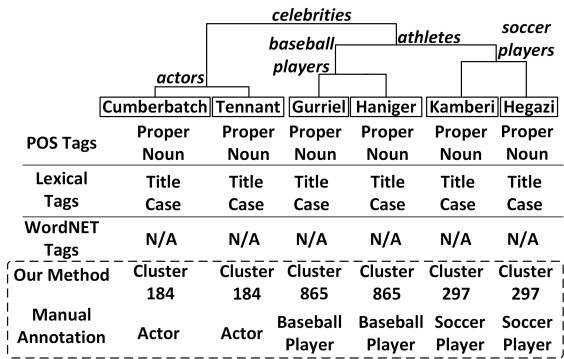

| | Cumberbatch | Tennant | Gurriel | Haniger | Kamberi | Hegazi |
|---|---|---|---|---|---|---|
| POS Tags | Proper Noun | Proper Noun | Proper Noun | Proper Noun | Proper Noun | Proper Noun |
| Lexical Tags | Title Case | Title Case | Title Case | Title Case | Title Case | Title Case |
| WordNET Tags | N/A | N/A | N/A | N/A | N/A | N/A |
| Our Method | Cluster 184 | Cluster 184 | Cluster 865 | Cluster 865 | Cluster 297 | Cluster 297 |
| Manual Annotation | Actor | Actor | Baseball Player | Baseball Player | Soccer Player | Soccer Player |

Figure 1: Illustration of a hierarchical concept tree. Our new concepts represent novel aspects that are not found in the pre-defined concepts.

$K = 1000$ with an aim to avoid many large clusters of size more than 1000 tokens (under-clustering) and a large number of small clusters having less than five word types (over-clustering)[2].

## 3.2 DATA PREPARATION

The choice of a dataset is an important factor to be considered when generating latent concepts. Ideally, one would like to form latent clusters over a large collection with diverse domains. However, clustering a large number of points is computationally and memory intensive. In the case of agglomerative hierarchical clustering using Ward linkage, the memory complexity is $O(n^2)$, which translates to about 400GB of runtime memory for 200k input vectors. A potential workaround is to use a different linkage algorithm like single or average linkage, but each of these come with their own caveats regarding assumptions around how the space is structured and how noisy the inputs are.

We address these problems by using a subset of a large dataset of News 2018 WMT[3] (359M tokens). We randomly select 250k sentences from the dataset ($\approx$5 million tokens). We further limit the number of tokens to $\approx$210k by discarding singletons, closed-class words, and tokens with a frequency higher than 10. For every token, we extract its contextualized representations and embedding layer representation using the 12-layered `BERT-base-cased` model (Devlin et al., 2019) in the following way: First, we tokenize sentences using the Moses tokenizer[4] and pass them through the standard pipeline of BERT as implemented in HuggingFace.[5] We extract layer-wise representations of every token. If the BERT tokenizer splits an input token into subwords or multiple tokens, we mean pool over them to create an embedding of the input token. For every layer, we cluster tokens using their contextualized representations.

## 3.3 CONCEPT LABELS

We define a hierarchical concept-tagset, starting with the core language properties i.e., semantics, parts-of-speech, lexical, syntax, etc. For each cluster, we assign the core properties that are represented by that cluster, and further enrich the label with finer hierarchical information. It is worth noting that the core language properties are not mutually exclusive i.e., a cluster can belong to more than one core properties at the same time. Consider a group of words mentioning *first name* of the *football players* of the *German* team and all of the names occur at the *start of a sentence*, the following series of tags will be assigned to the cluster: `semantic:origin:Germany`, `semantic:entertainment:sport:football`, `semantic:namedentity:person:firstname`, `syntax:position:firstword`. Here, we preserve the hierarchy at various levels such as, sport, person name, origin, etc., which can be used to combine clusters to analyze a larger group.

---

[2]We experimented with Elbow and Silhouette but they did not show reliable results.
[3]`http://data.statmt.org/news-crawl/en/`
[4]`https://pypi.org/project/mosestokenizer/`
[5]`https://huggingface.co`

### 3.4 ANNOTATION TASK

Following our definition of concepts, we formulated the annotation task using the two questions below:

> Q1: Is the cluster meaningful?
> Q2: Can the two clusters be combined to form a meaningful group?

We show annotators the contextual information (i.e., corresponding sentences) of the words for each of these questions. The annotator can answer them as *(i)* yes, *(ii)* no, and *(iii)* don't know or can't judge. Q1 mainly involves identifying if the cluster is meaningful, according to our definition of a concept (Section 2). Based on the *yes* answer of Q1, annotators' task is to either introduce a new concept label where it belongs or classify it into an appropriate existing concept label. Note that at the initial phase of the annotation, the existing concept label set was empty. As the annotators annotate and finalize the concept labels, we accumulate them into the existing concept label set and provide them in the next round of annotations, in order to facilitate and speed-up the annotations. In Q2, we ask the annotators to identify whether two sibling clusters can be also combined to form a meaningful super-cluster. Our motivation here is to keep the hierarchical information that BERT is capturing intact. For example, a cluster capturing Rock music bands in the US can be combined with its sibling that groups Rock music bands in the UK to form a Rock music bands cluster. Similarly, based on the *yes* answer of Q2, the annotators are required to assign a concept label for the super-cluster. The annotation guidelines combined with the examples and screenshots of the annotation platform is provided in Appendix B.

The annotators are asked to maintain hierarchy while annotating e.g., a cluster of ice hockey will be annotated as `semantic:entertainment:sport:ice_hockey` since the words are semantically grouped and the group belongs to sports, which is a subcategory of entertainment. In case of more than one possible properties that form a cluster, we annotate it with both properties e.g., title case and country names where the former is a lexical property and the latter is a semantic property.

Since annotation is an expensive process, we carried out annotation of the final layer of BERT only. The concept annotation is performed by three annotators followed by a consolidation step to resolve disagreements if there are any. For the annotation task, we randomly selected 279 clusters out of 1000 clusters obtained during the clustering step (see Section 3.1).

## 4 ANNOTATED DATASET

### 4.1 INTER ANNOTATION AGREEMENT

We computed Fleiss $\kappa$ (Fleiss et al., 2013) and found the average value for Q1 and Q2 to be 0.61 and 0.64, respectively, which is *substantial* in the $\kappa$ measurement (Landis & Koch, 1977). The detail on $\kappa$ measures and agreement computation using other metrics are reported in Appendix (Table 2).

### 4.2 STATISTICS

For Q1, we found 243 (87.1%) meaningful clusters and 36 (12.9%) non-meaningful clusters. For Q2, we found that 142 (75.9%) clusters out of 187 clusters can form a meaningful cluster when combined with their siblings. The annotation process resulted in 174 unique concept labels from the 279 clusters (See Appendix B.3 for the complete list of labels). The high number of concept labels is due to the fine-grained distinctions and multi-facet nature of the clusters where a cluster can get more than one labels. The label set consists of 11 lexical labels, 10 morphological labels, 152 semantic labels and one syntactic label. The semantic labels form the richest hierarchy where the next level of hierarchy consists of 42 labels such as: entertainment, sports, geo-politics, etc.

### 4.3 EXAMPLES OF CONCEPT HIERARCHY

In this Section, we describe the concept hierarchy that we have achieved through our annotation process. Figure 1 illustrates an example of a resulting tree found by hierarchical clustering. Each node is a concept candidate. We capture different levels of fine-grained meanings of word groups, which do not exist in pre-defined concepts. For example, a SEM or an NE tagger would mark a name as person, but our discovery of latent concepts show

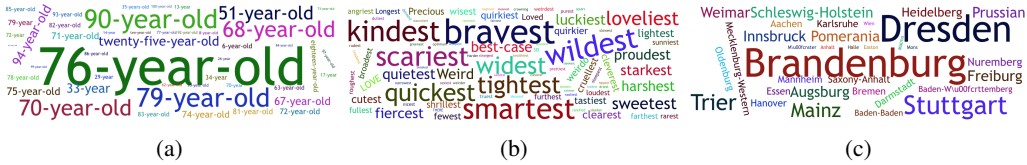

(a)                                        (b)                                        (c)

Figure 2: Example Concepts. 2a shows a cluster that has a lexical property (*hyphenation*) and a morphological property (*adjective*). Similarly, 2b shares a common suffix *est* and additionally possesses the morphological phenomenon, superlative adjectives. Finally, a large number of clusters were grouped based on the fine-grained semantic information. For example, the cluster in 2c is a named entity cluster specific to places in Germany.

that BERT uses a finer hierarchy by further classifying them as "baseball players", "soccer players", and "athletes". Our annotation process preserves this information and provides hierarchical labels such as `semantic:named-entity:person:celebrity:actor` and `semantic:named-entity:person:athletes:soccer`. Due to the multi-faceted nature of the latent clusters, many of them are annotated with more than one concept label. In the case of the clusters shown in Figure 1, they have a common lexical property (title case) and hence will also get the label `LEX:case:title` during annotation. Figure 2 shows word clouds of a few annotated clusters along with their descriptions.[6]

## 5 ANALYSIS AND FINDINGS

### 5.1 QUALITATIVE ANALYSIS

Our annotated dataset unveils insightful analysis of the latent concepts learned within BERT representations. Figure 3 presents a few examples of the discovered concepts. The two clusters of decimal numbers (Figure 3a, 3b) look quite similar, but are semantically different. The former represents numbers appearing as percentages e.g., `9.6%` or `2.4 percent` and the latter captures monetary figures e.g., `9.6 billion Euros` or `2.4 million dollars`. We found these two clusters to be sibling, which shows that BERT is learning a morpho-semantic hierarchy where it grouped all the decimal numbers together (morphologically) and then further made a semantic distinction by dividing them into percentages and monetary figures. Such a subtle difference in the usage of decimal numbers shows the importance of using fine-grained concepts for interpretation. For example, numbers are generally analyzed as one concept (`CD` tag in POS) which may be less informative or even misleading given that BERT treats them in a variety of different ways. Figure 3c shows an example of a cluster where the model captures an era of time, specifically 18 and 19 hundreds. Learning such information may help the model to learn relation between a particular era and the events occurring in that time period. Similarly, we found BERT learning semantic distinction of animal kingdom and separated animals into land, sea and flying groups (see Appendix B.3). These informative categories support the use of pre-trained models as a knowledge base (Petroni et al., 2019).

We further observed that BERT learns aspects specific to the training data. Figure 3d shows a cluster of words where female roles (Moms, Mothers, Granny, Aunt) are grouped together with job roles such as (housekeeper, Maid, Nanny). The contextual information in the sentences where these words appear, do not explicate whether the person is a male or a female, but based on the data used for the training of BERT, it has associated these roles to females. For example, the sentence "*Welfare staff 's' good **housekeeping** praised for uncovering disability benefits error*" is a general sentence. However, the contextualized representation of *housekeeping* in this sentence associates it to a female-related concept. Similarly, in another example BERT clustered names based on demography even though the context in which the word is used, does not explicitly provide such information. An example is the sentence, "*Bookings: Dzagoev, Schennikov, **Musa** , Akinfeev*" where the name *Musa* is used in a diverse context but it is associated to a cluster of specific demography (Arab region). While it is natural to form clusters based on specific data aspects, we argue that the use of such concepts while making a general prediction raises concerns of bias and fairness. For example, a loan prediction application should not rely on the background of the name of the applicant.

---

[6]The size of a word represents it's relative frequency in the cluster.

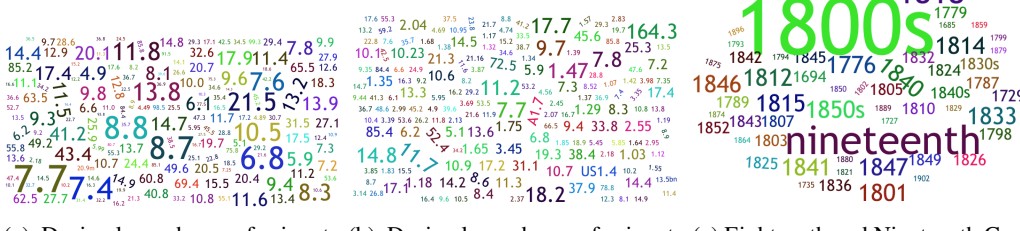

(a) Decimal numbers referring to percentages.

(b) Decimal numbers referring to value of money.

(c) Eighteenth and Nineteenth Century.

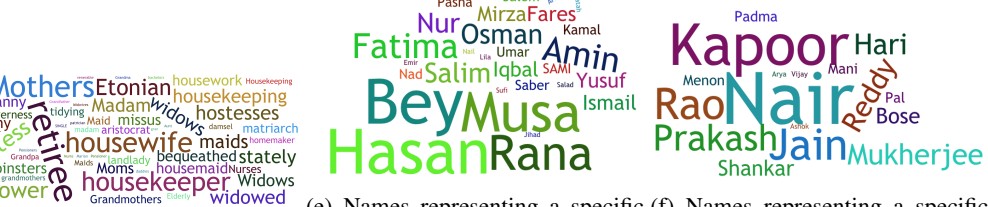

(d) Feminine roles.

(e) Names representing a specific demography

(f) Names representing a specific demography

Figure 3: Examples of latent concepts learned in BERT

## 5.2 ALIGNMENT WITH PRE-DEFINED CONCEPTS

A number of works on interpreting DNNs study to what extent various linguistic concepts such as parts-of-speech tags, semantic tags, etc are captured within the learned representations. This is in view of one school of thought which believes that an NLP model needs to learn linguistic concepts in order to perform well and to generalize (Marasović, 2018). For example, the holy grail in machine translation is that a proficient model ought to be aware of word morphology, grammatical structure, and semantics to do well (Vauquois, 1968; Jones et al., 2012). In this section, we explored how well the latent concepts in BERT align with the pre-defined linguistic concepts.

We compared the resulting clusters with the following pre-defined concepts: parts of speech (POS) tags (Marcus et al., 1993), semantic tags (SEM) using the Parallel Meaning Bank data (Abzianidze et al., 2017), CCG supertagging using CCGBank (Hockenmaier, 2006), syntactic chunking (Chunking) using CoNLL 2000 shared task dataset (Tjong Kim Sang & Buchholz, 2000), WordNet (Miller, 1995) and LIWC psycholinguistic-based tags (Pennebaker et al., 2001). We also introduced lexical and syntactic concepts such as casing, ngram matches, affixes, first and last word in a sentence etc. Appendix D provides the complete list of pre-defined concepts. For POS, SEM, Chunking, and CCG tagging, we trained a BERT-based classifier using the training data of each task and tagged the words in the selected News dataset. For WordNet and LIWC, we directly used their lexicon.

We consider a latent concept to align with a corresponding pre-defined concept if 90% of the cluster's tokens have been annotated with that pre-defined concept. For example, Figure 3a and 3b are assigned a tag `POS:CD` (parts-of-speech:cardinal numbers). Table 1 shows the statistics on the number of clusters matched with the pre-defined concepts. Of the 1000 clusters, we found the linguistic concepts in POS to have maximum alignment (30%) with the latent concepts in BERT, followed by Casing (23%). The remaining pre-defined concepts including various linguistic categories aligned with fewer than 10% of the latent concepts. We further explored the parallelism between the hierarchy of latent and pre-defined linguistic concepts using POS tags. We merged the POS tags into 9 coarse tags and aligned them with the latent concepts. The number of latent concepts aligned to these coarse concepts increased to 50%, showing that the model has learned coarser POS concepts and does not strictly follow the same fine-grained hierarchy.

## 5.3 LAYER-WISE COMPARISON OF CONCEPTS

**How do the latent concepts evolve in the network?** To answer this question, we need to manually annotate per-layer latent concepts and do a cross-layer comparison. However, as manual annotation is an expensive process, we used pre-defined concepts as a proxy to annotate the concepts and

| | Lexical | | | Morphology and Semantics | | | | Syntactic | | |
|---|---|---|---|---|---|---|---|---|---|---|
| Concepts | Ngram | Suffix | Casing | POS | SEM | LIWC | WordNet | CCG | Chunk | FW |
| Matches | 20 | 5 | 229 | 297 | 96 | 15 | 39 | 87 | 63 | 35 |
| | (2.0%) | (0.5%) | (23%) | (30%) | (10%) | (1.5%) | (3.9%) | (8.7%) | (6.3%) | (3.5%) |

Table 1: Alignment of BERT concepts (Layer 12) with the pre-defined concepts

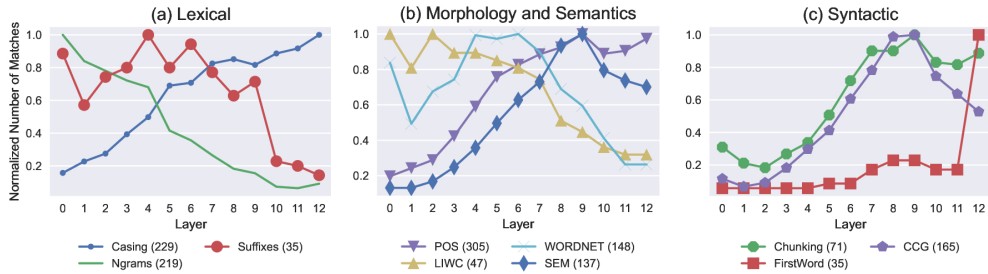

Figure 4: Layer-wise alignment of pre-defined concepts in BERT. The vertical axis represents the number of aligned clusters for a given concept, normalized by the maximum across all layers. Numbers inside parenthesis show the maximum alignment for each concept.

compare across layers. Figure 4 summarizes the layer-wise matching of pre-defined concepts with the latent concepts. Interestingly, the layers differ significantly in terms of the concepts they capture. In Figure 4a, the embedding layer is dominated by the concepts based on common ngram affixes.[7] The number of ngram-based shallow lexical clusters consistently decreases in the subsequent layers. The suffixes concept showed a different trend, where the information peaks at the lower-middle layers and then drops in the higher layers. The dominance of ngrams-based concepts in the lower layers is due to the subword segmentation. The noticeable presence in the middle layers might be due to the fact that suffixes often maintain grammatical information unlike other ngrams. For example, the ngram "ome" does not have a linguistic connotation, but suffix "ful" converts a noun into an adjective. Interestingly, casing information is least represented in the lower layers and is consistently improved for higher layers. Casing in English has a morphological role, marking a named entity, which is an essential knowledge to preserve at the higher layers.

The patterns using the classical NLP tasks: POS, SEM, CCG and Chunking all showed similar trends (See Figure 4b and 4c). The concepts were predominantly captured at the middle and higher layers, with minimal representation at the lower layers. WordNet and LIWC, which are cognitive and psychology based grouping of words respectively, showed a rather different trend. These concepts had a higher match at the embedding layer and a consistent drop after the middle layers. In other words, the higher and last layers of BERT are less aligned with the cognitive and psychology based grouping of words, and these concepts are better represented at the embedding and lower layers. We manually investigate a few clusters from the lower layers that match with WordNet and LIWC. We found that these clusters group words based on similar meaning e.g. all words related to "family relations" or "directions in space" form a cluster (See Appendix C for more examples). Lastly, Figure 4c also shows that the FirstWord position information is predominantly learned at the last layers only. This is somewhat unexpected, given that no clusters were based on this concept in the embedding layer, despite it having an explicitly position component.

These trends reflect the evolution of knowledge across the network: lower layers encode the lexical and meaning-related knowledge and with the inclusion of context in the higher layers, the encoded concepts evolve into representing linguistic hierarchy. These findings are in line with Mamou et al. (2020) where they found the emergence of linguistic manifolds across layer depth.

## 5.4 UNALIGNED CLUSTERS

In this section, we discuss the concepts that were not matched with the pre-defined concepts. We have divided them into 3 categories: i) latent concepts that can be explained via composition of

---

[7]For every cluster we pick a matching ngram with highest frequency match. Longest ngram is used when tie-breaker is required. We ignore unigrams.

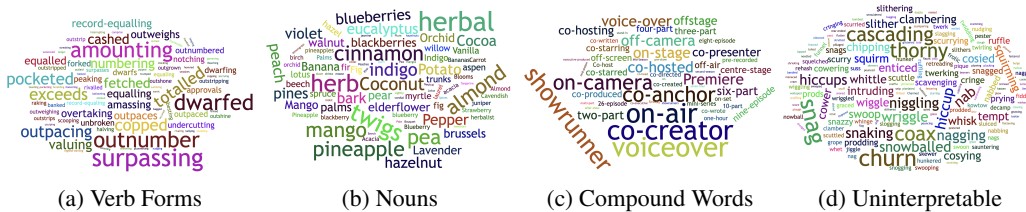

| (a) Verb Forms | (b) Nouns | (c) Compound Words | (d) Uninterpretable |

Figure 5: Unaligned Concepts

two pre-defined concepts, ii) concepts that cannot be explained via auto-labels but are meaningful and interpretable, iii) concepts that are uninterpretable. Figure 5a and 5b are examples of the first category mentioned above. The former describes a concept composed of different verb forms. It does not fully align with any of the fine-grained verb categories but can be deemed as a verb cluster at coarser level. Similarly the latter shows a cluster containing singular and plural nouns (described by POS:NN and POS:NNS) and WordNet concepts (WN:noun:food and WN:noun:plants) and can be aligned with a coarser noun category. Figure 5c shows a cluster that does not match with any human-defined concept but are interpretable in that the cluster is composed of compound words. Lastly in Figure 5d, we show a cluster that we found to be uninterpretable.

## 6 BERT CONCEPTNET DATASET (BCN)

In addition to the manually labeled dataset, we also developed a large dataset, BCN, using a weakly supervised approach. The motivation of creating such a dataset is to provide a reasonable size dataset that can be effectively use for interpretation studies and other tasks. The idea is to expand the number of tokens for every concept cluster by finding new token occurrences that belong to the same cluster. There are several possible ways to develop such a dataset. Here, we explored two different approaches. *First:* we compute the centroid for each cluster and find the cluster that is closest to the representation of any given new token. However, this did not result in accurate cluster assignment because of the non-spherical and irregularly shaped clusters. *Second:* we developed a logistic regression classifier (as shown in Appendix A, Algorithm 2) using our manually annotated dataset to assign new tokens to given concept clusters. We found the second approach better than the former in terms of token to cluster assignment. We trained the model using 90% of the concept clusters and evaluate its performance on the remaining 10% concept clusters (held-out set). The classifier achieved a precision score of 75% on the held-out set. In order to achieve higher precision, we introduce a threshold $t = 0.97$ on the confidence of the predicted cluster id, assigning new tokens to particular clusters only when the confidence of the classifier is higher than the threshold. This enables better precision in cluster assignment to the new tokens, at the expense of discarding some potentially good assignments, which can be offset by using more data for prediction.

Using the trained model we labeled all the tokens from the 2 million random sentences from the unused News 2018 dataset (see Section 3). The resulting dataset: BCN is a unique multi-faceted resource consisting of 174 concept labels with a total of 997,195 annotated instances. The average number of annotated instances per concept label are 5,731. The utility of this resource is not limited to interpretation, but can serve as a valuable fine-grained dataset for the NLP community at large. The hierarchy present in the concept labels provides flexibility to use data with various granularity levels. Appendix F provides the detailed statistics for BCN.

**Case Study** In order to showcase the efficacy of BCN as a useful resource in interpretation, we present a case-study based on neuron-level interpretation in this section. The goal of neuron interpretation is to discover a set of neurons responsible for any given concept (Sajjad et al., 2021). Most of the work in this direction Lakretz et al. (2019); Durrani et al. (2020); Torroba Hennigen et al. (2020) identified neurons responsible for concepts belonging to the core NLP tasks such as morphology, syntax and semantics. In this work, we found that BERT learns finer categories compared to the ones defined in the core NLP tasks. For example, BCN consists of a large number of fine-grained proper noun categories, each representing a unique facet such as `proper-noun:person:specific-demography`. This enables identifying neurons learning very fine-grained properties, thus provides a more informed interpretation of models.

In a preliminary experiment, we performed neuron analysis using the PER (person) concept of SEM and the fine-grained concept of *Muslim names* (Figure 3e). We then compared the amount of neurons required for each concept. For the SEM dataset, we used the standard splits (Appendix D). For the fine-grained concept, the BCN dataset consists of 2748 *Muslim name* instances. We randomly selected an equal amount of negative class instances and split the data into 80% train, 10% development and 10% test sets. We used Linguistic Correlation Analysis (Dalvi et al., 2019a), using the NeuroX toolkit Dalvi et al. (2019b) and identified the minimum number of neurons responsible for each concept. We found that only 19 neurons are enough to represent the fine-grained concept as opposed to 74 neurons for the PER concept of SEM, showing that the BCN dataset enables selection of specialized neurons responsible for very specific aspects of the language. The discovery of such specialized neurons also facilitate various applications such as model control (Bau et al., 2019; Gu et al., 2021; Suau et al., 2020), domain adaptation (Gu et al., 2021) and debiasing.

## 7 RELATED WORK

The interpretation of DNNs in NLP has been largely dominated by the post-hoc model analysis (Belinkov et al., 2017a; Liu et al., 2019a; Tenney et al., 2019a) where the primary question is to identify which linguistic concepts are encoded in the representation. Researchers have analyzed a range of concepts varying from low-level shallow concepts such as word position and sentence length to linguistically motivated concepts such as morphology (Vylomova et al., 2016; Dalvi et al., 2017; Durrani et al., 2019), syntax (Shi et al., 2016; Linzen et al., 2016) and semantics (Qian et al., 2016; Belinkov et al., 2017b) and provide insights into what concepts are learned and encoded in the representation. One of the shortcomings of these studies is their reliance on pre-defined concepts for interpretation. Consequently, novel concepts that models learn about the language are largely undiscovered.

Recently Michael et al. (2020) analyzed latent concepts learned in pre-trained models using a binary classification task to induce latent concepts relevant to the task and showed the presence of linguistically motivated and novel concepts in the representation. Mamou et al. (2020) analyzed representations of pre-trained models using mean-field theoretic manifold analysis and showed the emergence of linguistic structure across layer depth. Similar to these two studies, we aim to analyze latent concepts learned in the representation. However different from them, we analyze raw representations independent of a supervised task. The reliance on a supervision task effects the type of latent concepts found and may not fully reflect the latent concepts encoded in the raw representation. Moreover, the post-hoc classifiers require a careful evaluation to mitigate the concerns regarding the power of the probe (Hewitt & Liang, 2019; Zhang & Bowman, 2018). In this work, we address these limitations by analyzing latent concepts learned in BERT in an unsupervised fashion.

## 8 CONCLUSION

In this work, we have provided a comprehensive analysis of latent concepts learned in the BERT model. Our analysis revealed various notable findings such as, i) the model learns novel concepts which do not strictly adhere to any pre-defined linguistic groups, ii) various latent concepts are composed of multiple independent properties, such as proper-nouns and first word in a sentence, iii) lower layers specialize in learning shallow lexical concepts while the middle and higher layers have a better representation of core linguistic and semantic concepts. We also released a novel BERT-centric concept dataset that not only facilitates future interpretation studies, but will also serve as an annotated dataset similar to POS, WordNet and SEM. Different from the classical NLP datasets, it provides a unique multi-facet set of concepts.

While this work has led to plenty of insight into the inner workings of BERT and led to a novel resource, increasing the diversity of the observed concepts is an important direction of future work. Working around the memory and algorithm constraints will allow the creation of a more varied concept set. Phrase-level concepts is yet another direction that can be explored further. Finally, a few works on analyzing word representations (both static and contextual) like Reif et al. (2019) and Ethayarajh (2019) have alluded that taking the principal components into account may lead to a more controlled analysis. Incorporating this line of work may yield useful insights into the type of discovered latent clusters.

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

APPENDIX

## A    CLUSTERING ALGORITHM AND BCN ALGORITHM

---
**Algorithm 1** Learning Latent Concept with Clustering

---
**Input**: $\overrightarrow{y}^l$ = word representation of all running words

**Parameter**: $K$ = No. of output clusters

  1: **for** each word $w_i$ **do**
  2:     assign $w_i$ to cluster $c_i$
  3: **end for**
  4: **while** No. of clusters $\neq K$ **do**
  5:     **for** each cluster pair $c_i, c_{i'}$ **do**
  6:       $d_{i,i'}$ = inner-cluster difference of combined cluster $c_i$ and $c_{i'}$
  7:     **end for**
  8:     $c_j, c_{j'}$ = cluster pair with minimum value of $d$
  9:     merge clusters $c_j$ and $c_{j'}$
10: **end while**

---

---
**Algorithm 2** Concept Prediction with Logistic regression

---
**Input**: $X_{train}$ = word representations for train data, $Y_{train}$ = cluster id from Algorithm 1, $X_{test}$ = word representations for test data,

**Parameter**: $t$ = probability threshold

  1: $c$ = unique cluster ids from $Y_{train}$
  2: $\mathbb{M}$ = train Logistic Regression model on $X_{train}$ and $Y_{train}$
  3: **for** each $x \in X_{test}$ **do**
  4:     $p$ = predict $K$ probabilities for each cluster using $\mathbb{M}$ and input $x$
  5:     $i = \arg\max p$
  6:     **if** $p_i >= t$ **then**
  7:       assign $x$ to cluster id $c_i$
  8:     **end if**
  9: **end for**

---

## B    ANNOTATION DETAILS

For the annotation, we prepared detailed instructions to guide the annotators, which they followed during the annotation tasks. Our annotation consists of two questions: *(i)* Q1: Is the cluster meaningful?, *(ii)* Q2: Can the two clusters be combined to form a meaningful group?. The word cluster is represented in a form of a word cloud, where the frequency of the word in the data represents a relative size of the word in the word cloud. To understand the context of each word in the cluster we also facilitated annotators with associated sentences from the dataset. For the annotation, we provided specific instructions with examples for each question. With the second question, the idea is to understand whether two sibling clusters can be combined to form a meaningful super-cluster. For the second question, two clusters are siblings, which we automatically identified from our hierarchical clustering model. We showed the second question only if a sibling of the main cluster (cluster presented in the first question) was available or identified by the clustering algorithm. Hence, annotators did not see the second question in all annotations. Below we provided instructions which we accompanied with the annotation platform in a form of a tutorial.

### B.1    ANNOTATION INSTRUCTIONS

The annotation task was to look first look at a group of words (i.e., representing as a cluster) and answer the first question. Then, look at two groups of words to answer the second question.

### Q1: IS THE CLUSTER MEANINGFUL?

A word group is meaningful if it contains *semantically*, *syntactically*, or *lexically* similar words. The example in Figure 6 has only numbers with two digits, hence, this is a meaningful group. The labels for this question include the following:

1. **Yes** (if represents a meaningful cluster)

2. **No** (if it does not represent any meaningful cluster)

3. **Don't know or can't judge** (if it does not have enough information to make a judgment. It is recommended to categorize the word groups using this label when the word group is not understandable at all.)

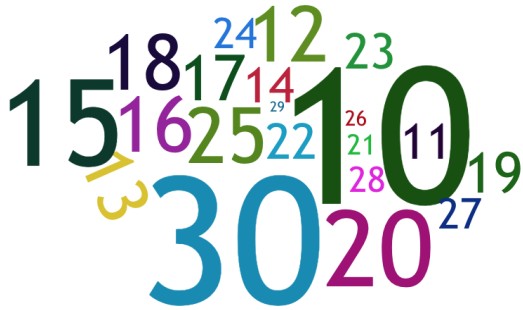

Figure 6: Example of a group of tokens representing a meaningful cluster.

If the answer to this question is *Yes*, then the task would be to assign a name to the word cluster using one or more words. While assigning the name, it is important to maintain the hierarchy. For example, for the above word group in Figure 6, we can assign a name: semantic:number. This needs to be written in the text box right below this question. While deciding the name of the cluster, the priority has to be to focus on semantics first, syntax second, and followed by any other meaningful aspects. While annotating it is also important to consider *(i)* their relative frequency of the tokens in the cluster, which is clearly visible in the word cluster, *(ii)* context in a sentence where the word appears in.

Q2: CAN THE TWO CLUSTERS BE COMBINED TO FORM A MEANINGFUL GROUP?

For this question, two clusters are shown and the task is to see if they can form a meaningful super-cluster after combining. In these two clusters, the left one is the same cluster annotated for the first question. The answer (i.e., labels) of this question is similar to Q1: *Yes*, *No* and *Don't know or can't judge*. Depending on the answer to this question the task is to provide a meaningful name similar to Q1.

In Figure 7 and 8 we provide screenshots for Q1 and Q2, respectively.

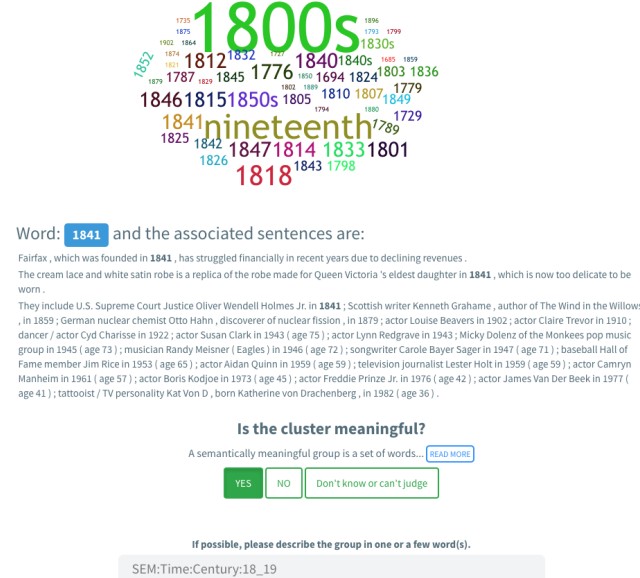

Figure 7: An example of the annotation interface for Q1.

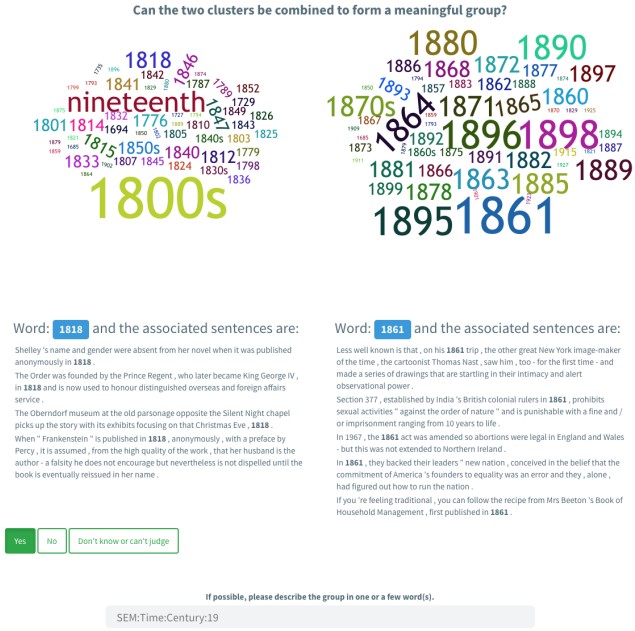

Figure 8: An example of the annotation interface for Q2.

| Agree. Pair | Fleiss $\kappa$ | K. $alpha$ | Avg. Agr. |
|---|---|---|---|
| **Q1** | | | |
| A1 - C | 0.604 | 0.604 | 0.915 |
| A2 - C | 0.622 | 0.623 | 0.921 |
| A3 - C | 0.604 | 0.604 | 0.915 |
| **Avg** | 0.610 | 0.610 | 0.917 |
| **Q2** | | | |
| A1 - C | 0.702 | 0.700 | 0.853 |
| A2 - C | 0.625 | 0.621 | 0.814 |
| A3 - C | 0.602 | 0.592 | 0.797 |
| **Avg** | 0.643 | 0.638 | 0.821 |

Table 2: Inter-annotator agreement using Fleiss Kappa ($\kappa$). *A* refers to annotator, and *C* refers to consolidation.

## B.2 ANNOTATION AGREEMENT

We computed annotation agreement using Fleiss $\kappa$ Fleiss et al. (2013), Krippendorff's $\alpha$ Krippendorff (1970) and average observed agreement Fleiss et al. (2013). In Table 2, we present the annotation agreement with different approaches.

## B.3 CONCEPT LABELS

Out annotation process resultant in 183 concept labels. In Table 3, 4 and 5 we report LEX, POS and SEM concepts labels, respectively.

## C ANALYSIS

Figure 9 shows example concepts. Figure 10 shows examples of LIWC and WordNet concepts found in lowe layers.

| | LEX | POS |
|---|---|---|
| LEX:abbreviation | LEX:suffix:ties | POS:adjective |
| LEX:abbreviation_and_acronym | LEX:unicode | POS:adjective:superlative |
| LEX:acronym | LEX:suffix:ly | POS:adverb |
| LEX:case:title_case | LEX:suffix:ion | POS:compound |
| LEX:dots | POS:proper-noun | POS:noun |
| LEX:hyphenated | POS:verb:present:third_person_singular | POS:noun:singular |
| LEX:prefix:in | POS:verb | POS:number |

Table 3: Concept labels: **LEX** and **POS**.

| SEM | | |
|---|---|---|
| SEM:action | SEM:entertainment:music:US_rock-bands_and_musicians | SEM:honour_system |
| SEM:action:body:face | SEM:entertainment:sport | SEM:language |
| SEM:activism:demonstration | SEM:entertainment:sport:american_football | SEM:language:foreign_word |
| SEM:agent | SEM:entertainment:sport:american_football:game_play | SEM:measurement:height |
| SEM:agent:passerby | SEM:entertainment:sport:baseball | SEM:measurement:imperial |
| SEM:animal:land_animal | SEM:entertainment:sport:basketball | SEM:measurement:length |
| SEM:animal:sea_animal | SEM:entertainment:sport:club_name | SEM:measurement:length_and_weight |
| SEM:art:physical | SEM:entertainment:sport:cricket | SEM:media:social_media |
| SEM:attribute:human | SEM:entertainment:sport:football | SEM:media:tv:channel |
| SEM:baby_related | SEM:entertainment:sport:game_score | SEM:medicine:drug_related |
| SEM:brand | SEM:entertainment:sport:game_time | SEM:medicine:medical_condition |
| SEM:crime:assault | SEM:entertainment:sport:ice_hockey | SEM:metaphysical:death_related |
| SEM:crime:tool | SEM:entertainment:sport:player_name | SEM:mining_and_energy |
| SEM:defense:army | SEM:entertainment:sport:rugby | SEM:named_entity |
| SEM:defense:army:title | SEM:entertainment:sport:team_name | SEM:named_entity:location |
| SEM:defense:guns | SEM:entertainment:sport:team_selection | SEM:named_entity:location:city |
| SEM:demography | SEM:entertainment:sport:winter_sport | SEM:named_entity:location:city_and_county |
| SEM:demography:age | SEM:entertainment:vacation:outdoor | SEM:named_entity:location:city_and_state |
| SEM:demography:age:young | SEM:fashion | SEM:named_entity:location:county |
| SEM:demography:muslim_name | SEM:fashion:apparel | SEM:named_entity:organization |
| SEM:donation_and_recovery | SEM:fashion:clothes | SEM:named_entity:person |
| SEM:entertainment | SEM:financial | SEM:named_entity:person:first_name |
| SEM:entertainment:actors | SEM:financial:money_figure | SEM:named_entity:person:initial |
| SEM:entertainment:fictional | SEM:financial:stock | SEM:named_entity:person:last_name |
| SEM:entertainment:fictional:games_of_thrones | SEM:food:wine_related | SEM:negative |
| SEM:entertainment:film:hollywood | SEM:food_and_plant | SEM:number |
| SEM:entertainment:film_and_tv | SEM:gender:feminine | SEM:number:alpha_numeric |
| SEM:entertainment:gaming | SEM:geopolitics | SEM:number:cardinal:spelled |
| SEM:entertainment:music | SEM:group:student | SEM:number:floating_point |
| SEM:entertainment:music:classical | SEM:historic:medieval | SEM:number:less_than_hundred |

Table 4: Concept labels: **SEM.**

| SEM | |
|---|---|
| SEM:number:one_hundred_to_five_hundred | SEM:origin:north_america:mexico |
| SEM:number:ordinal | SEM:origin:north_america:usa:california |
| SEM:number:percentage | SEM:origin:polynesia |
| SEM:number:quantity:concrete | SEM:origin:portuguese |
| SEM:number:quantity:vague | SEM:origin:south_america:brazil |
| SEM:number:whole | SEM:origin:spanish_italian |
| SEM:organization:government | SEM:polarity |
| SEM:organization:government:foreign_affairs | SEM:renovation-related |
| SEM:organization:social_ngo | SEM:science:biology:micro_biology |
| SEM:origin:africa | SEM:science:chemistry |
| SEM:origin:asia:arab | SEM:science:chemistry:material |
| SEM:origin:asia:east_asian | SEM:science:geology |
| SEM:origin:asia:indochina:french_colony | SEM:sea_related |
| SEM:origin:asia:myanmar | SEM:service-industry |
| SEM:origin:asia:south_east_asian | SEM:technology |
| SEM:origin:asia:subcontinent | SEM:technology:communication |
| SEM:origin:europe | SEM:technology:electronic |
| SEM:origin:europe:belgium_and_netherlands | SEM:technology:electronic:storage-devices:removable |
| SEM:origin:europe:dutch | SEM:time |
| SEM:origin:europe:france | SEM:time:anniversary_descriptors |
| SEM:origin:europe:germany | SEM:time:century:18th_and_19th |
| SEM:origin:europe:germany_related | SEM:time:months_of_year |
| SEM:origin:europe:greece_and_italy | SEM:time:timeframe |
| SEM:origin:europe:north_europe | SEM:transportation:way |
| SEM:origin:europe:portugal | SEM:transportation:way:road |
| SEM:origin:europe:sweden | SEM:unnamed-entity:role |
| SEM:origin:europe:switzerland | SEM:vehicle |
| SEM:origin:europe:uk | SEM:vehicle:bike |
| SEM:origin:french | SEM:weather |
| SEM:origin:latin_america | SEM:weather:disaster |
| SEM:origin:latin_america_related | SYN:position:first_word |
| SEM:origin:north_america:canada | |

Table 5: Concept labels: **SEM.**

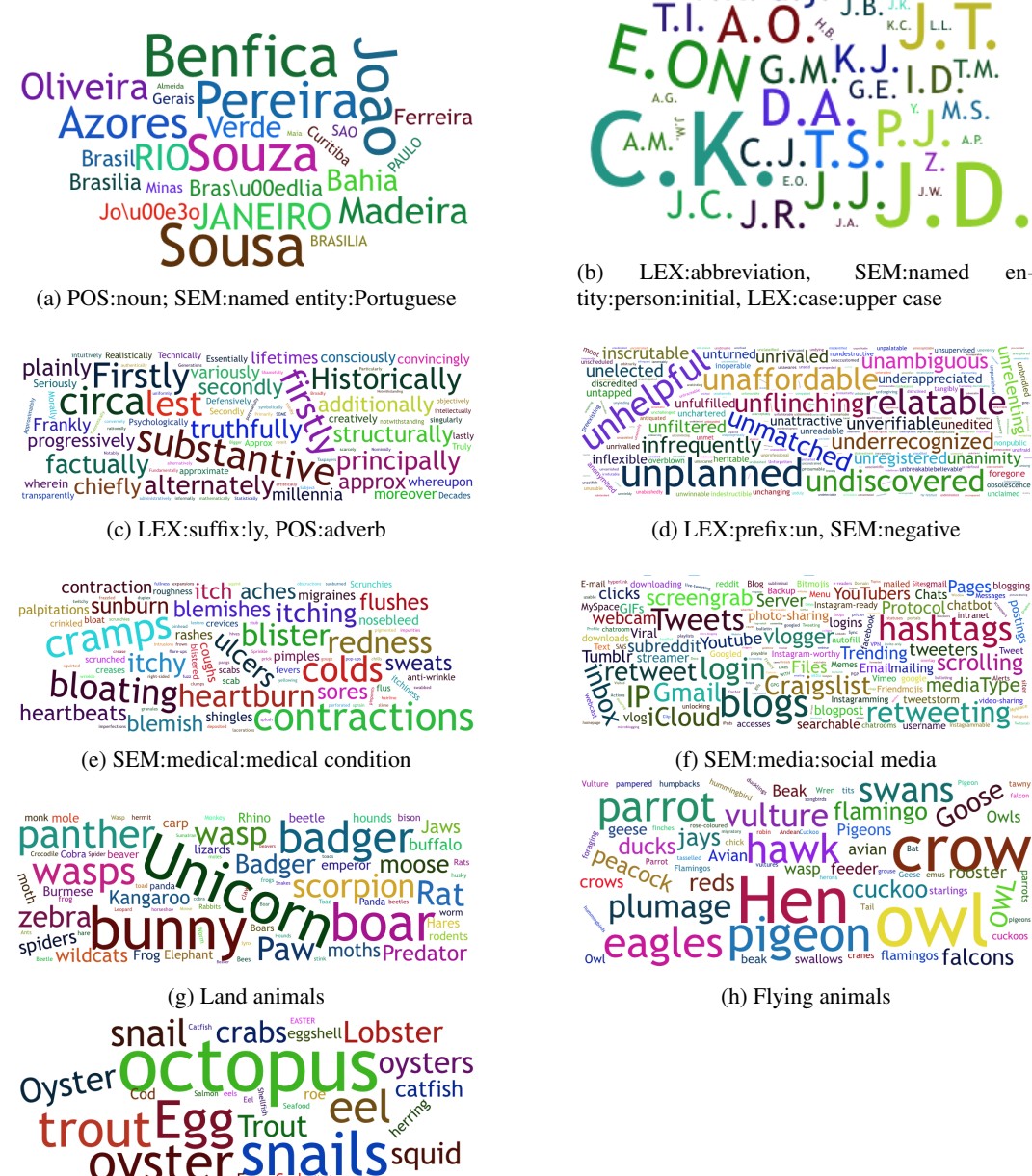

(a) POS:noun; SEM:named entity:Portuguese

(b) LEX:abbreviation, SEM:named entity:person:initial, LEX:case:upper case

(c) LEX:suffix:ly, POS:adverb

(d) LEX:prefix:un, SEM:negative

(e) SEM:medical:medical condition

(f) SEM:media:social media

(g) Land animals

(h) Flying animals

(i) Sea animals

Figure 9: Example Concepts

# D  PRE-DEFINED CONCEPTS INFORMATION

## D.1  PRE-DEFINED CONCEPTS

Table 6 provides a list of all pre-defined concepts used in this work.

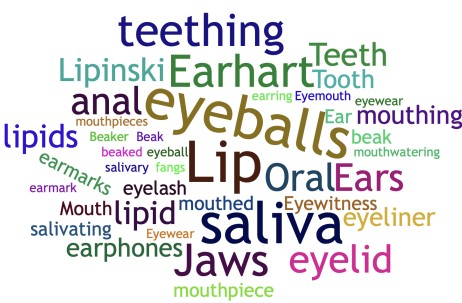

(a) Layer 1: LIWC:bio

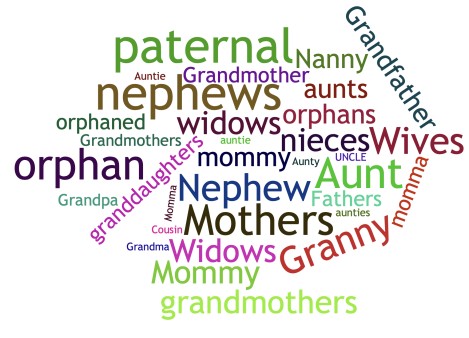

(b) Layer 1: LIWC:social, WordNet:Noun:Person

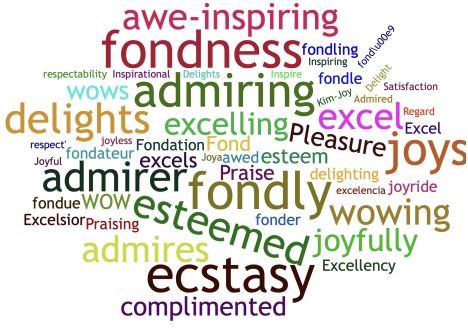

(c) Layer 2: LIWC:affect

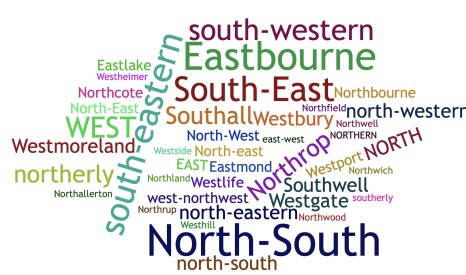

(d) Layer 2: LIWC:space

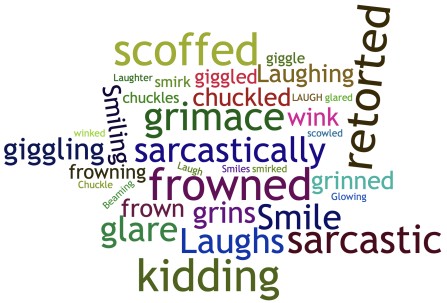

(e) Layer 3: LIWC:affect

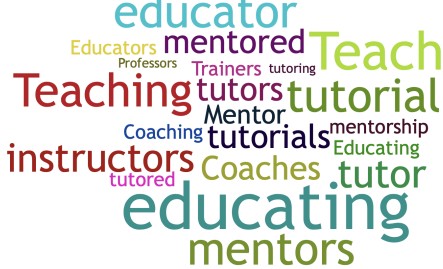

(f) Layer 3: LIWC:work

Figure 10: LIWC and WordNet concepts found at lower layers.

| Type | Concepts |
|---|---|
| Lexical | Ngram, Prefix, Suffix, Casing |
| Morphology | Parts of speech |
| Semantics | Lexical semantic tags, LIWC, WordNet, Named entity tags |
| Syntactic | CCG, Chunking, First word, Last word |

Table 6: Pre-defined concepts

## D.2 DATA AND EXPERIMENTAL SETUP

We used standard splits for training, development and test data for the 4 linguistic tasks (POS, SEM, Chunking and CCG super tagging). The splits to preprocess the data are available through git

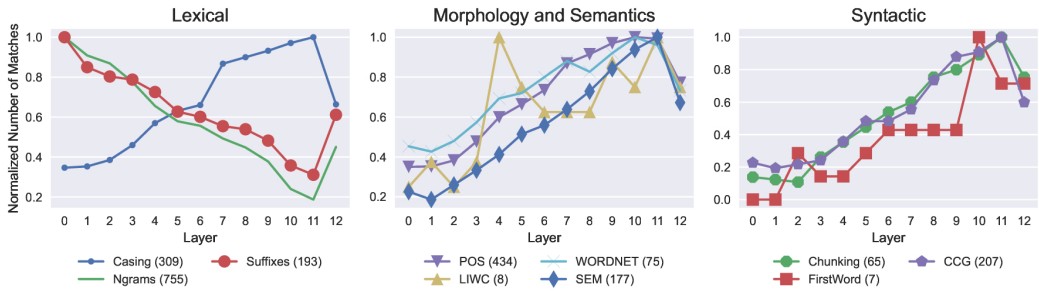

Figure 11: RoBERTa (numbers inside brackets show the maximum match across all layers)

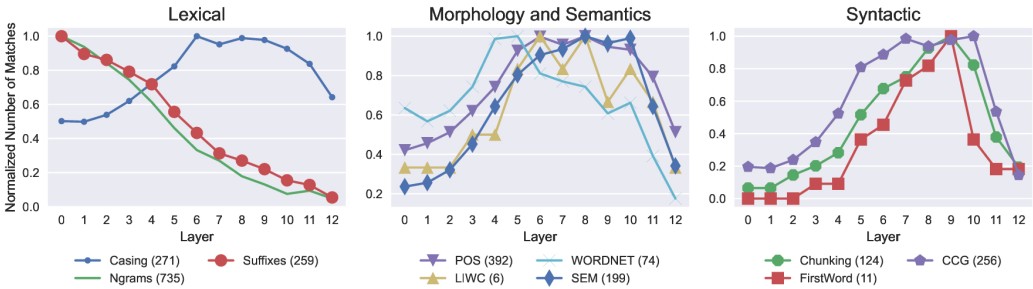

Figure 12: XLNet (numbers inside brackets show the maximum match across all layers)

repository[8] released with Liu et al. (2019a). See Table 7 for statistics. We obtained the understudied pre-trained models from the authors of the paper, through personal communication.

| Task | Train | Dev | Test | Tags |
|---|---|---|---|---|
| POS | 36557 | 1802 | 1963 | 44 |
| SEM | 36928 | 5301 | 10600 | 73 |
| Chunking | 8881 | 1843 | 2011 | 22 |
| CCG | 39101 | 1908 | 2404 | 1272 |

Table 7: Data statistics (number of sentences) on training, development and test sets using in the experiments and the number of tags to be predicted

## E  COMPARING PRE-TRAINED MODELS

**How do models compare with respect to the pre-defined concepts?** Similar to BERT, we created layer-wise latent clusters using RoBERTa (Liu et al., 2019b) and XLNet (Yang et al., 2019). We aligned the clusters with the pre-defined concepts. Figure 11 and 12 shows the results. The overall trend of the models learning shallow concepts in lower layers and linguistic concepts in the middle-higher layers is consistent across all models. A few exceptions are: suffixes have a consistently decreasing pattern for XLNet and RoBERTa, the FirstWord information improved for every higher layer till the last few layers. Moreover, XLNet showed a significant drop in matches with the linguistic tasks for the last layers. The last layers of the models are optimized for the objective function (Kovaleva et al., 2019) and there the drop in matches for most of the concepts reflects the existence of novel task-specific concepts which may not be well aligned with the human-defined concepts.

The number of matches as shown in brackets in Figure 4, 11 and 12 provides a relative comparison across models and pre-defined concepts. The RoBERTa and XLNet found substantially more

---

[8]https://github.com/nelson-liu/contextual-repr-analysis

| Concept | # Token | # Type |
|---|---|---|
| LEX:abbreviation_and_acronym | 4,921 | 948 |
| LEX:abbreviation, SEM:medicine:medical_condition | 760 | 221 |
| LEX:abbreviation, SEM:named_entity:organization, POS:proper-noun | 8,615 | 1,194 |
| LEX:abbreviation, SEM:named_entity:person:initial, LEX:case:upper_case | 2,039 | 177 |
| LEX:case:title_case, POS:proper-noun, SEM:entertainment:sport, SEM:named_entity:person:last_name | 429 | 29 |
| LEX:case:title_case, POS:proper-noun, SEM:named_entity:person, SEM:named_entity:person:last_name, SEM:origin:europe:germany | 2,299 | 1,033 |
| LEX:case:title_case, POS:proper-noun, SEM:named_entity:person:first_name | 518 | 195 |
| LEX:case:title_case, POS:proper-noun, SEM:named_entity:person:last_name | 5,911 | 1,164 |
| LEX:case:title_case, POS:proper-noun, SEM:origin:europe:uk, SEM:named_entity | 3,032 | 1,585 |
| LEX:case:title_case, SEM:entertainment, POS:proper-noun, SEM:named_entity:person:first_name, | 1,208 | 545 |
| LEX:case:title_case, SEM:named_entity:location, POS:proper-noun | 2,622 | 167 |
| LEX:case:title_case, SEM:named_entity:location, SEM:origin:north_america:canada, POS:proper-noun | 2,597 | 236 |
| LEX:case:title_case, SEM:named_entity:person, SEM:entertainment:actors, POS:proper-noun | 4,154 | 589 |
| LEX:case:title_case, SEM:named_entity:person, SEM:entertainment:fictional:games_of_thrones, POS:proper-noun | 227 | 64 |
| LEX:case:title_case, SEM:named_entity:person, SEM:origin:asia:east_asian, POS:proper-noun | 2,875 | 184 |
| LEX:case:title_case, SEM:named_entity:person:last_name, SEM:entertainment:sport | 3,502 | 908 |
| LEX:case:title_case, SEM:origin:polynesia, POS:proper-noun, SEM:named_entity | 3,936 | 82 |
| LEX:case:title_case, SYN:position:first_word, POS:proper-noun, SEM:named_entity:person | 3,481 | 1,772 |
| LEX:case:upper_case, SYN:position:first_word, SEM:named_entity:location:city, POS:proper-noun | 3,279 | 289 |
| LEX:dots | 22,547 | 31 |
| LEX:hyphenated, POS:adjective, POS:compound, SEM:geopolitics | 2,309 | 205 |
| LEX:hyphenated, SEM:demography:age, LEX:suffix:-year-old, POS:adjective | 7,847 | 251 |
| LEX:hyphenated:words | 15,922 | 1,060 |
| LEX:prefix:un, SEM:negative | 4,058 | 652 |
| LEX:suffix:est, POS:adjective:superlative | 19,014 | 366 |
| LEX:suffix:ly, POS:adverb | 10,039 | 706 |
| LEX:suffix:ly, POS:adverb | 6,187 | 871 |
| LEX:suffix:ly, POS:adverb | 75,166 | 1,089 |
| LEX:suffix:ly, POS:adverb, | 1,943 | 240 |
| LEX:suffix:ly, POS:adverb, SEM:number:ordinal, | 489 | 15 |
| LEX:unicode | 596 | 411 |
| POS:adjective, SEM:attribute:human | 2,474 | 286 |
| POS:number, SEM:number:quantity:vague, SEM:number:one_hundred_to_five_hundred, SEM:entertainment:sport:cricket, SEM:entertainment:sport:game_score | 558 | 262 |
| POS:proper-noun, LEX:case:title_case, SEM:entertainment:sport:football, SEM:named_entity:person, SEM:entertainment:sport:player_name, | 1,367 | 323 |
| POS:proper-noun, LEX:case:title_case, SEM:named_entity, SEM:origin:latin_america_related | 2,603 | 968 |
| POS:proper-noun, LEX:case:title_case, SEM:named_entity:location, SEM:origin:europe:belgium_and_netherlands | 1,740 | 83 |
| POS:proper-noun, LEX:case:title_case, SEM:named_entity:person, SEM:entertainment:sport:player_name, | 1,051 | 166 |
| POS:proper-noun, SEM:named_entity, SEM:media:tv:channel | 3,359 | 55 |
| POS:proper-noun, SEM:named_entity, SEM:origin:europe:france | 2,165 | 392 |
| POS:proper-noun, SEM:named_entity, SEM:origin:europe:portugal | 1,571 | 27 |
| POS:proper-noun, SEM:named_entity, SEM:origin:south_america:brazil | 3,524 | 50 |
| POS:proper-noun, SEM:named_entity:location, SEM:origin:europe:france | 617 | 78 |
| POS:proper-noun, SEM:named_entity:person, SEM:demography:muslim_name, LEX:case:title_case | 2,748 | 106 |
| POS:proper-noun, SEM:origin:europe:germany, SEM:named_entity:location, LEX:case:title_case | 1,951 | 85 |
| POS:verb | 7,763 | 1,227 |
| POS:verb | 893 | 310 |
| POS:verb, SEM:action, SEM:negative | 1,858 | 427 |
| POS:verb:present:third_person_singular | 32,984 | 1,870 |

Table 8: Concept labels (LEX, POS) with token and type.

matches with most of the classical NLP tasks compared to BERT. The difference is much more pronounced in the case of POS with RoBERTa have the highest match. This may reflect that the latent concepts of RoBERTa follow a closer hierarchy to POS compared to other models. On contrary, for WordNet and LIWC BERT showed substantially better alignment than the other two models.

## F    BCN DETAILS

In Table 8, 9 and 10, we provide concept label with the number of token and type for LEX, POS and SEM.

| Concept Label | # Token | # Type |
|---|---|---|
| SEM:action, SEM:negative, POS:verb | 2,908 | 364 |
| SEM:action:body:face | 1,553 | 101 |
| SEM:agent, POS:noun:singular, | 4,491 | 483 |
| SEM:agent, POS:noun:singular, | 12,227 | 348 |
| SEM:agent:passerby, POS:noun | 330 | 14 |
| SEM:animal:land_animal | 2,121 | 419 |
| SEM:animal:land_animal, | 1,705 | 129 |
| SEM:animal:land_animal, | 1,783 | 83 |
| SEM:animal:sea_animal | 1,920 | 57 |
| SEM:art:physical | 5,041 | 647 |
| SEM:baby_related | 2,184 | 181 |
| SEM:crime:assault, | 2,707 | 275 |
| SEM:crime:tool, | 892 | 34 |
| SEM:defense:army, | 6,619 | 331 |
| SEM:defense:army:title, LEX:case:title_case, POS:proper-noun, | 255 | 21 |
| SEM:defense:guns | 9,803 | 516 |
| SEM:demography, POS:adjective | 19,150 | 461 |
| SEM:demography:age:young | 27,929 | 274 |
| SEM:donation_and_recovery, LEX:case:title_case, SEM:named_entity, | 3,774 | 499 |
| SEM:donation_and_recovery, LEX:case:title_case, SEM:named_entity, | 2,642 | 147 |
| SEM:entertainment:fictional | 412 | 17 |
| SEM:entertainment:fictional, SEM:entertainment:film_and_tv | 2,236 | 548 |
| SEM:entertainment:fictional, WORDNET:noun.person, | 114 | 5 |
| SEM:entertainment:film_and_tv, | 6,104 | 562 |
| SEM:entertainment:film_and_tv, | 5,007 | 196 |
| SEM:entertainment:gaming | 2,690 | 555 |
| SEM:entertainment:music | 6,982 | 710 |
| SEM:entertainment:music:classical | 3,768 | 390 |
| SEM:entertainment:music:US_rock-bands_and_musicians, LEX:case:title_case | 2,939 | 436 |
| SEM:entertainment:sport | 3,616 | 154 |
| SEM:entertainment:sport | 9,734 | 704 |
| SEM:entertainment:sport | 15,491 | 2,178 |
| SEM:entertainment:sport, | 1,979 | 484 |
| SEM:entertainment:sport, | 15,973 | 1,094 |
| SEM:entertainment:sport, | 5,619 | 962 |
| SEM:entertainment:sport, SEM:origin:europe:uk | 18,300 | 232 |
| SEM:entertainment:sport, SEM:vehicle:bike | 2,335 | 76 |
| SEM:entertainment:sport:american_football:game_play | 14,621 | 388 |
| SEM:entertainment:sport:baseball, SEM:entertainment:sport:game_score, POS:number | 1,835 | 383 |
| SEM:entertainment:sport:team_selection | 12,919 | 815 |
| SEM:entertainment:sport:winter_sport | 6,112 | 405 |
| SEM:entertainment:vacation:outdoor | 2,600 | 149 |
| SEM:fashion:apparel | 707 | 108 |
| SEM:fashion:clothes | 1,882 | 376 |
| SEM:financial, POS:noun, | 7,003 | 308 |
| SEM:financial, POS:noun, | 907 | 47 |
| SEM:financial:stock | 995 | 201 |
| SEM:food_and_plant | 3,601 | 256 |
| SEM:food:wine_related | 494 | 20 |
| SEM:gender:feminine, SEM:unnamed-entity:role | 5,627 | 278 |
| SEM:group:student | 596 | 60 |
| SEM:historic:medieval, | 2,011 | 431 |
| SEM:historic:medieval, SEM:origin:europe | 621 | 85 |
| SEM:honour_system, SEM:origin:europe:uk, | 1,124 | 128 |
| SEM:language, | 8,166 | 336 |
| SEM:language, | 607 | 101 |
| SEM:language:foreign_word, SEM:origin:europe:dutch | 1,038 | 394 |
| SEM:language:foreign_word, SEM:origin:french | 3,321 | 916 |
| SEM:language:foreign_word, SEM:origin:french, SEM:named_entity | 2,993 | 867 |
| SEM:measurement:height, LEX:hyphenated | 1,397 | 275 |
| SEM:measurement:length | 3,112 | 673 |
| SEM:measurement:length_and_weight, SEM:measurement:imperial | 960 | 156 |
| SEM:media:social_media | 42,108 | 975 |
| SEM:media:social_media, SEM:technology:communication, | 10,192 | 116 |
| SEM:medicine:drug_related | 1,625 | 92 |
| SEM:medicine:medical_condition | 1,340 | 261 |
| SEM:metaphysical:death_related | 4,977 | 164 |
| SEM:mining_and_energy | 4,072 | 279 |
| SEM:named_entity, LEX:case:title_case | 2,784 | 861 |
| SEM:named_entity, POS:proper-noun, SEM:vehicle:bike, | 1,353 | 139 |
| SEM:named_entity, SEM:entertainment:sport:basketball, POS:proper-noun | 4,870 | 79 |
| SEM:named_entity, SEM:entertainment:sport:ice_hockey, SEM:entertainment:sport:club_name, LEX:case:title_case | 2,570 | 101 |
| SEM:named_entity, SEM:entertainment:sport:rugby, SEM:entertainment:sport:team_name | 223 | 9 |
| SEM:named_entity, SEM:origin:asia:east_asian, POS:proper-noun, SEM:named_entity:person, LEX:case:title_case, | 835 | 258 |
| SEM:named_entity, SEM:origin:asia:myanmar, POS:proper-noun, LEX:case:title_case | 2,758 | 172 |

Table 9: Concept labels **SEM** with token and type.

| Concept Label | # Token | # Type |
|---|---|---|
| SEM:named_entity, SEM:origin:europe:germany_related, LEX:case:title_case, | 2,958 | 725 |
| SEM:named_entity, SEM:origin:europe:north_europe, POS:proper-noun | 1,159 | 331 |
| SEM:named_entity, SEM:origin:europe:sweden, POS:proper-noun | 2,881 | 423 |
| SEM:named_entity:location, LEX:case:title_case, SEM:origin:asia:south_east_asia | 330 | 17 |
| SEM:named_entity:location, LEX:case:title_case, SEM:origin:latin_america | 152 | 7 |
| SEM:named_entity:location, SEM:origin:europe:greece_and_italy, LEX:case:title_case, SEM:historic:medieval | 2,512 | 407 |
| SEM:named_entity:location:city_and_county, LEX:case:title_case | 359 | 12 |
| SEM:named_entity:location:city_and_county, SEM:origin:north_america:usa:california | 19,267 | 297 |
| SEM:named_entity:location:city_and_state, SEM:origin:north_america:mexico, POS:proper-noun, LEX:case:title_case | 1,946 | 123 |
| SEM:named_entity:location:county, LEX:suffix:shire | 407 | 15 |
| SEM:named_entity:organization, POS:proper-noun, LEX:acronym | 2,379 | 266 |
| SEM:named_entity:person, LEX:case:title_case, POS:proper-noun, SEM:entertainment | 1,842 | 306 |
| SEM:named_entity:person, POS:proper-noun, LEX:case:title_case | 1,159 | 783 |
| SEM:named_entity:person, POS:proper-noun, LEX:hyphenated, SEM:named_entity:person:last_name, LEX:case:title_case | 3,107 | 630 |
| SEM:named_entity:person, SEM:origin:asia:arab | 2,532 | 851 |
| SEM:named_entity:person, SEM:origin:asia:indochina:french_colony, LEX:case:title_case, POS:proper-noun | 1,680 | 259 |
| SEM:named_entity:person:first_name, LEX:case:title_case, SYN:position:first_word | 1,024 | 456 |
| SEM:named_entity:person:first_name, POS:proper-noun, SEM:entertainment, LEX:case:title_case, | 479 | 189 |
| SEM:named_entity:person:last_name, LEX:case:title_case, SEM:origin:asia:subcontinent, POS:proper-noun | 3,515 | 1,491 |
| SEM:named_entity:person:last_name, POS:proper-noun, LEX:case:title_case | 4,190 | 1,153 |
| SEM:named_entity:person:last_name, POS:proper-noun, LEX:case:title_case, SEM:origin:europe:germany_related, | 1,056 | 380 |
| SEM:named_entity:person:last_name, SEM:entertainment, LEX:case:title_case, POS:proper-noun | 3,884 | 696 |
| SEM:named_entity:person:last_name, SEM:entertainment:film:hollywood, LEX:case:title_case, POS:proper-noun, SEM:entertainment | 930 | 300 |
| SEM:named_entity:person:last_name, SEM:entertainment:sport, POS:proper-noun, LEX:case:title_case, | 792 | 216 |
| SEM:named_entity:person:last_name, SEM:origin:europe:germany, POS:proper-noun, LEX:case:title_case, | 1,400 | 612 |
| SEM:named_entity:person:last_name, SEM:origin:spanish_italian, POS:proper-noun, LEX:case:title_case | 707 | 251 |
| SEM:named_entity—SEM:origin:europe:switzerland | 2,290 | 80 |
| SEM:negative, POS:noun | 4,141 | 365 |
| SEM:number, SEM:entertainment:sport:game_score, SEM:entertainment:sport:american_football, LEX:hyphenated, POS:number | 1,860 | 471 |
| SEM:number:alpha_numeric | 1,222 | 387 |
| SEM:number:cardinal:spelled, SEM:number:less_than_hundred | 8,203 | 131 |
| SEM:number:cardinal:spelled, SEM:number:less_than_hundred, LEX:hyphenated, LIWC:funct, | 634 | 136 |
| SEM:number:floating_point, POS:number, SEM:number:quantity:concrete, SEM:number:percentage | 5,001 | 829 |
| SEM:number:floating_point, SEM:number, SEM:financial:money_figure, SEM:number:quantity:vague, LEX:position:after_currency_symbol, LEX:position:before_suffix_-illion | 4,568 | 798 |
| SEM:number:ordinal, SEM:time, SEM:entertainment:sport:game_time, SEM:entertainment:sport:football | 599 | 103 |
| SEM:number:quantity:vague, LEX:suffix:0, SEM:number:whole, POS:number, | 10,689 | 2,997 |
| SEM:number:quantity:vague, LEX:suffix:0, SEM:number:whole, POS:number, | 14,242 | 1,430 |
| SEM:organization:government, SEM:agent, | 14,744 | 516 |
| SEM:organization:government:foreign_affairs | 3,139 | 46 |
| SEM:organization:social_ngo | 5,132 | 151 |
| SEM:origin:africa, LEX:case:title_case, SEM:named_entity | 8,115 | 257 |
| SEM:origin:asia:east_asian, SEM:named_entity, POS:proper-noun, | 2,564 | 84 |
| SEM:origin:europe:uk, POS:proper-noun | 3,734 | 108 |
| SEM:polarity, LEX:hyphenated, LEX:prefix:anti_and_pro | 6,884 | 662 |
| SEM:renovation-related, LEX:prefix:re | 2,905 | 289 |
| SEM:science:biology:micro_biology | 2,105 | 403 |
| SEM:science:chemistry | 1,690 | 131 |
| SEM:science:chemistry, | 3,138 | 321 |
| SEM:science:chemistry:material | 604 | 155 |
| SEM:science:geology | 5,453 | 663 |
| SEM:sea_related | 16,583 | 287 |
| SEM:service-industry | 2,743 | 244 |
| SEM:technology | 4,480 | 149 |
| SEM:technology:communication | 4,138 | 463 |
| SEM:technology:communication, SEM:technology:electronic | 5,155 | 886 |
| SEM:technology:electronic:storage-devices:removable | 387 | 21 |
| SEM:time, | 21,715 | 526 |
| SEM:time:anniversary_descriptors | 1,629 | 25 |
| SEM:time:century:18th_and_19th | 1,041 | 204 |
| SEM:time:months_of_year | 16,041 | 37 |
| SEM:time:timeframe, LEX:suffix:ties | 1,062 | 20 |
| SEM:time:timeframe, POS:adjective, LEX:hyphenated | 8,471 | 623 |
| SEM:transportation:way, POS:noun, | 2,637 | 211 |
| SEM:transportation:way:road, POS:noun, | 1,722 | 72 |
| SEM:vehicle, SEM:named_entity, POS:proper-noun, SEM:brand | 811 | 103 |
| SEM:weather, | 861 | 20 |
| SEM:weather:disaster, | 29,698 | 1,631 |
| SYN:position:first_word, | 7,913 | 1,315 |
| SYN:position:first_word, | 2,252 | 365 |
| SYN:position:first_word, LEX:case:title_case, POS:proper-noun, SEM:named_entity:person | 1,627 | 342 |
| SYN:position:first_word, LEX:case:title_case, SEM:activism:demonstration | 4,572 | 518 |
| SYN:position:first_word, LEX:case:title_case, SEM:weather:disaster | 1,351 | 232 |
| SYN:position:first_word, POS:proper-noun, LEX:case:title_case, SEM:named_entity:person | 2,059 | 879 |
| SYN:position:first_word, POS:proper-noun, LEX:case:title_case, SEM:named_entity:person, SEM:origin:asia:east_asian, | 294 | 134 |
| SYN:position:first_word, SEM:named_entity | 3,842 | 1,863 |

Table 10: Concept labels **SEM** with token and type.

| Example of Sem tags | Words |
|---|---|
| SEM:action:negative | confine, cripple, decimate, demonised, disappoints, dislocation, disobeyed, disqualify, distort, downsized, jeopardise, hijacked, trashed, traumatizing, undercutting, underpins, underplayed, undervalue, spoilt, spouting, spurned, stifled |
| SEM:action:body:face | Gaze, LAUGH, Tears, admiring, blushing, chuckles, clap, frown, gasp, giggle, grimace, murmur, smirk, twinkle, wink |
| SEM:activism:demonstration | Aggression, Arguments, Armored, Attackers, Attacks, Backlash, Ceasefire, Clashes, Cops, Counterfeiting |
| SEM:animal:land_animal | Ants, Bees, Beetle, Cobra, Frog, Jaws, Leopard, Paw, Snakes, Spider, Sumatran, frog |
| SEM:animal:sea_animal | Salmon, Shellfish, Trout, catfish, crabs, eel, herring, octopus, oyster, oysters, snail, squid, trout |

Table 11: Example of Sem tags with associated words.

