# OpenReview forum: "Discovering Latent Concepts Learned in BERT"
_ICLR.cc/2022/Conference — ICLR 2022 Poster_

### Official Review · Reviewer_We8g · 2021-10-27

**Correctness:** 3
**Technical Novelty And Significance:** 3
**Empirical Novelty And Significance:** 2
**Recommendation:** 8
**Confidence:** 4

**Main Review:**

**Strengths**

- The topic studying "latent concepts" is interesting and potentially impactful.

- This paper involves a lot of empirical effort.
- The dataset would be beneficial for future researches.

**Weaknesses**

(W1) There appears to be a potential confound in annotation. Sec 3 loosely defines "concept" to be "a group of words that are meaningful". The annotation procedure seems to let humans to decide whether the words are meaningful based on the words themselves (instead of the contexts). Ethayarajh (2019) showed that the context matter a lot for contextualized representations (e.g., BERT). This annotation procedure appears to drop some useful information. E.g., what if the words themselves are not "meaningful", but the way they are used in contexts appear meaningful? Do the annotators mark them as "not meaningful"?

(W2) The generalizability of the finding is limited. The results reported in this paper appear specific to BERT. How would the clustering results compare to a static embedding (e.g., GloVe) and other contextualized representations (e.g., RoBERTa)? As far as I understand, the representation vectors cluster well because the meanings themselves follow hierarchical clusters, and that different DNN models can indeed embed their meanings fairly well -- BERT, especially, is one of the most powerful DNN models, so it is not surprising that it can do well here. Just showing "BERT clusters them well" does not add novel knowledge to what people already know. Comparing to other models might bring in more novel findings.

(W3) The reporting of the results needs to be improved. Page 6 last paragraph: "We compared the resulting clusters with the following pre-defined concepts: ... CCGBank, WordNet, LIWC": Some comparison results are listed in Table 1. What is the percentage of matches? E.g., Lexical:NGram contains 20 matches, but how many concepts are there in total? In addition, Table 1 only shows the "top-level" match counts (& of layer 12). Among the more fine-grained concepts, how many concepts are matched?

**Questions to the authors**

(Q1) On the choice of concept database. In 4.3 Concept labels: there seem to be some existing databases with hierarchical concepts. E.g., WordNet. Why do you hand-engineer a concept database instead of using WordNet? Compared to WordNet, the hierarchical concepts defined in this paper appears very simplistic -- WordNet contains 155,327 words (let's say each "concept label" contains 100 words, then they have "concept label" to the magnitude of 1k), while the database of this paper contains 183 concept labels.

(Q2) During the clustering, do the optimal number of clusters remain consistent for representations coming from different layers? (Am I right in assuming that Figure 2 only shows that of the final layer representation?)

(Q3) On the limitation of the data size in clustering. How about first assigning each word to a "large cluster" (e.g., by the part-of-speech), which should be very accurate, and then run the cluster algorithm within each "large cluster"? This might help circumvent the computation bottleneck.

**Summary Of The Paper:**

**Summary**

While a large number of previous papers evaluate DNN models on pre-defined concepts, this paper interprets the models (i.e., BERT) on latent concepts that are learned in an unsupervised manner. The analyses in this paper include: (1) Agglomerative hierarchical clustering from ~200k embedded tokens. (2) Comparing the clustered concepts to some hierarchical concept-tagsets. The authors find that lower layers learn many shallow lexical concepts, while higher layers learn more semantic relations. In addition, this paper releases a novel BERT concept dataset (BCD). With cross validation, the authors show that the BCD dataset has high consistency.

**Contributions**

1. Interpret BERT by analyzing latent concepts learned in the network
2. Collect and annotate a hierarchical concept dataset (BCD)

**Summary Of The Review:**

Interesting topic and substantial empirical effort. Methodology (clustering, dataset annotation, etc.) has potential problems. Reporting needs improvements. The generalizability and novelty of the overall finding are limited.

---

> ### Author Response · Authors · 2021-11-22
> **Thank you. We have answered the comments and have improved the paper accordingly.**
>
> ***Q: … a potential confound in annotation … Ethayarajh (2019) showed that the context matter a lot for contextualized representations (e.g., BERT). This annotation procedure appears to drop some useful information...***
>
> A: Sorry for the confusion, perhaps we were not very clear in the paper. The context is important for the correct annotation and we provided clear instructions to the annotators to use contextual information while labeling. Our annotation interface facilitates annotators to look at both words and their context (appearing in sentences) to decide if the words form a meaningful cluster.
>  Please see the following screencast to have a clear idea about how annotators can see the context.  https://drive.google.com/file/d/1ze67nL6F_flNzf1LEKXPCck4UWNuggFq/view?usp=sharing
> We found several clusters which were “not meaningful” if considered without context e.g. the “first word” clusters. They can not be annotated correctly in the absence of context.
>
> ***Q: The generalizability of the finding is limited... specific to BERT... How would the clustering results compare … does not add novel knowledge to what people already know… Comparing to other models might bring in more novel findings.***
>
> A: We have added results for RoBERTa and XLNet with predefined concepts (Appendix, Figure 13). Except for the last layers, we observed the layer-wise trends to be quite consistent across models. We have also added discussion on unaligned concepts (section 6.4) that highlights compositional concepts and novel concepts that are not covered by pre-defined concepts.
>
> Regarding the embedding of meaning, we found the emergence of linguistic concepts across layer depth. The model groups words based on similar meaning in the embedding layer and initial layers, and with the presence of context and abstraction, it forms groupings based on core-linguistic concepts such as POS, Chunking, etc. The meaning-based groups are largely context independent groups while the core-linguistic concepts are contextual in nature, thus they evolved in the higher layers. We have added discussion on these lines in Section 6.3.
>
> ***Q: The reporting of the results needs to be improved... in Table 1. What is the percentage of matches? Among the more fine-grained concepts, how many concepts are matched?***
>
> A: We have added the percentage of matches in the paper. For POS, we found a total of 15 tags that were aligned with the latent concepts. The relatively less number of tag coverage is an artifact of the selected data that does not contain various closed class words. We ignored closed-class words (prepositions, articles, interjections etc) because they are less interesting concepts to analyze.
>
> ***Q: On the choice of concept database. In 4.3 Concept labels: there seem to be some existing databases with hierarchical concepts. E.g., WordNet. Why do you hand-engineer a concept database instead of using WordNet?***
>
> A: We agree to use labels from the already existing databases. However, we did not find any of the available resources complete enough to cover a good percentage of the latent concepts. This is visible in the number of matches provided in Figure 5. More concretely consider WordNet,  it does not cover named entities or other fine-grained information such as sports player names, a hierarchy (hypername to hyponame) that only relates to babies, etc. Similarly, other pre-defined resources have limitations and they can not be used to precisely define a latent concept. In our approach, we utilize all existing pre-defined concepts to guide the labeling process.
>
> ***Q: During the clustering, do the optimal number of clusters remain consistent for representations coming from different layers?***
>
> A: Yes, you are right about figure2. We made an experimental simplification to keep the analysis consistent. It is possible that the optimal number of clusters will vary across layers. This is an interesting direction to explore in the future.
>
> ***Q: On the limitation of the data size in clustering. How about first assigning each word to a "large cluster" (e.g., by the part-of-speech), which should be very accurate, and then run the cluster algorithm within each "large cluster"? …***
>
> A: This is a great suggestion and worth experimenting with, however we hypothesize that it will bias the results towards the linguistic category used to form the larger cluster, and we have to be careful about the resulting analysis. For example, we observed that the lower layer latent concepts group words based on similar meaning and they don’t follow core-linguistic categories. With this approach, we won’t be able to identify such clusters. In this work, our focus is to analyze latent clusters without getting biased from a pre-defined concept.

---

> > ### Comment · Reviewer_We8g · 2021-11-22
> > **Thank you for the reply**
> >
> > The reply and the update address my concerns regarding contextualization, generalization to other models, and the choice of creating new datasets. These make this paper more solid (while I'm still not surprised that deep LMs can cluster the concepts well in an ontological manner). I will increase the score.

---

> > > ### Author Response · Authors · 2021-11-23
> > > **Thank you**
> > >
> > > Thank you for acknowledging our response and for raising the score.

---

### Official Review · Reviewer_yqwt · 2021-10-29

**Correctness:** 3
**Technical Novelty And Significance:** 2
**Empirical Novelty And Significance:** 2
**Recommendation:** 5
**Confidence:** 4

**Main Review:**

The paper reads well, and the authors have clearly presented some interesting clusters within BERT. The authors use human intuition to hierarchically label a selection of "concept clusters" discovered in BERT. In this way, concept exploration in BERT is not limited to pre-defined concepts, and the authors promise to make their compiled dataset (BCD) available for additional research. Their results really do reveal some interesting clusters, and the paper's message could be bolstered if they included some tool to easily explore the concept space learned in BERT outside of the results shown in figures in the paper.


**Weaknesses & Feedback:**

1. In the main paper, the authors fail to describe the model that they use for analysis. All they say is "We base our study on BERT" (S1p1). But there are many flavors of BERT trained on different datasets, and the released BERT Concept Dataset (BCD) will only correspond to this model that the authors analyzed. There is not even a proper citation to BERT in the introduction of the paper (though one does appear later).
2. In addition to not disclosing the model architecture used in the main paper, the authors do not disclose the tokenization scheme. The most common tokenization schemes for a case-sensitive BERT would likely not have the long hyphenations (e.g., "75-year-old"/"90-year-old") and all the German words (e.g., "Mecklenburg-Western") as single tokens in the dictionary. How would this technique combine tokens to represent a single concept?
3. Without architecture information, I am left wondering about their statement that "200k vectors each of size 768 require 400GB of CPU memory". Assuming 32bit floating point, this many vectors should only occupy 0.6GB of space. Are they talking about the model memory? Why wouldn't you batch the representations and save them to hard disk to cluster later? I am also confused by this limitation.
4. The annotation tool examples in the Appendix do not make it clear if annotators had access to the original context each token appeared in. For example, the word clouds in Fig 4(a,b) have little information in the cluster itself to inform an annotator that they one refers to percentage and the other to units of money. The annotation tool seems to show 3 or 4 different sentences that a *single token from the cluster* came from. How does that generalize to the other tokens in the cluster?
5. (minor) There is a lot of speculation in the analysis as to how these findings could help, but I do not find the argument compelling. The authors could improve their narrative by elaborating on some of their claims. For example:
    - Learning about the existence of a cluster where the model captures an era of time "may help the model to learn relation between a particular era and the events occurring in that time period" (S6.1p6).
    - "A clothing recommendation system may use such information to customize suggestions towards a culture or a religion" (S6.1p6). I find this a questionable application that relies on biases learned in the language of a region/demographic.
    - The authors show no summary of the results in  S7.2p9 "Case Study", and most of this section is spent persuading readers that such analysis could be helpful. It also fails to state clearly how their "neuron analysis" could be replicated.
6. The annotation analysis performed in this paper is limited in that it only explores the representations in the final layer in BERT.
7. I have a few complaints about the word cloud visualization. The size of each token reflects the frequency of that token (footnote 2), but this is strongly biased by the total frequency of that token in the sub-sampled dataset. Perhaps more meaningful would be if the size of the tokens reflected the proximity to the centroid of the cluster.

**Summary Of The Paper:**

The paper performs a clustering analysis on the contextual representations of BERT on a subset of the News 2018 WMT corpus. These clusters are manually tagged by 3 annotators to connect these clusters to human concepts. The authors additionally release a dataset with these concepts labeled.

**Summary Of The Review:**

In summary, I find the main paper lacking in sufficient disclosure on methods, and I do not believe the techniques or findings of the paper to be sufficiently novel, though I can see that the dataset would be useful. I still vote reject for ICLR 2022.

---

> ### Author Response · Authors · 2021-11-22
> **Thank you. We have answered the comments and have improved the paper accordingly.**
>
> ***Q1: … the authors fail to describe the model that they use for analysis …***
>
> A: Thank you for pointing out the missing information. We added the details about the model used and the citation to BERT in the introduction. We used 12-layered BERT-base-cased model for our experiments as providing by huggingface.
>
> ***Q1: … the released BERT Concept Dataset (BCD) will only correspond to this model that the authors analyzed ...***
>
> A: This is correct to some degree. However, we would like to clarify that the utility of our work is not limited to BERT and can easily be extended for other models. In order to extend the analysis to other models, we can automatically map labels of BCD to other models, say RoBERTa, using a distant supervision approach. Moreover, BCD can be used as a supervised set (similar to POS and SEM tagging) to interpret any pretrained model against fine-grained categories, similar to the use case shown in the paper.
>
> ***Q2: … authors do not disclose the tokenization scheme …***
>
> A: Thank you for pointing this out. We have added it in the paper. We used a 12-layered BERT-base-cased model for our study with its respective subword tokenization as provided on the huggingface website. The BERT tokenizer segments words in tokens (“75-year-old” is tokenized as “75 - year - old”) and subword units (“fumbling” to “fu ##mbling”). We mean pool over the representations of the tokenized words/subwords to get a single representation of the original token. Since we pre-tokenized sentences before input to BERT, the punctuations are represented as a separate token in the initial input to BERT.
>
> ***Q3: … "200k vectors each of size 768 require 400GB of CPU memory". Assuming 32bit floating point, this many vectors should only occupy 0.6GB of space ...***
>
> A: The memory constraint does not come from storing the activations themselves, but from the memory complexity of the underlying algorithm. Since we are using Hierarchical clustering with ward linkage, the off-the-shelf implementation computes pairwise distances between all points in space (O(n^2) space complexity). For 200,000 points, this implies a matrix of 200,000 * 200,000 elements, which roughly amounts to 320 GB with double precision. The scikit implementation also uses memory for other things internally (maintaining the hierarchical dendrogram etc), and around 400GB is what we have seen experimentally on the dataset used in the paper.
>
> ***Q4: ... if annotators had access to the original context each token appeared in ...***
>
> A: Context plays an important role in deciding whether a cluster is meaningful or not. In the annotation platform, we provided top N sentences of each word. The feature is implemented as a mouse-hover over the words in the cluster. Please see the following video to get an idea of the annotators view:  \url{https://drive.google.com/file/d/1ze67nL6F_flNzf1LEKXPCck4UWNuggFq/view?usp=sharing}
>
> ***Q5: Minor comments***
>
> A: Thank you for pointing this out. We have improved the writing to clarify some of the claims.
>
> ***Q6: The annotation analysis … is limited … explores the representations in the final layer in BERT.***
>
> A: Annotation is expensive. We chose to start with the final layer since it is closest to the objective function, and is a good reflection of knowledge used for downstream tasks [1,2]. We would also like to highlight that this is the first study in this direction and our goal is to test our hypothesis, methodology and annotation setup for a single layer first. We will extend the analysis to other layers and models in the future work. In addition, we made our annotation scheme, annotation platform and all resources publicly available for the community to further extend the research in this direction.
> [1] Elzar et al (2020): Amnesic Probing: Behavioral Explanation with Amnesic Counterfactuals
> [2] Lakretz et al (2019): The emergence of number and syntax units in LSTM language models
>
> ***Q7: … complaints about the word cloud visualization...***
>
> A: The size of a word reflects that most of its contextualized representations are clustered in that particular cluster. This information is important to be considered when deciding about the label of the cluster. For example, if a word “A” appears 10 times in a cluster and the word “B” appears only single time in the cluster, the word “A” should have more influence in deciding the label of the cluster. Regarding using the centroid of a cluster, this is a good suggestion for algorithms that result in spherical clusters e.g. k-means. The hierarchical clustering results in non-spherical clusters and considering the mean in this case would be a mis-representation of the cluster.

---

> > ### Comment · Reviewer_yqwt · 2021-11-23
> > **Thank you for your response**
> >
> > I thank the authors for their response and for clarifying the BERT model, tokenization tricks, and visualization method used in their study.
> >
> > ***...the utility of our work is not limited to BERT and can be easily extended to other models...***
> >
> > I would argue that the approach of hierarchical clustering is not the right choice to study concepts learned in large language models given its memory-hungry nature. The authors themselves admitted this as a limitation to their approach. This method, paired with the requirement for human annotation, is incredibly expensive and actually a large barrier to extending this work on other models. After all, <25% of the 1000 clusters were annotated by hand.
> >
> > The demoed tool shows a relatively polished interface with autocomplete for some existing labels that can help the annotators create consistent and meaningful clusters. The tool, though, was not the main focus of the paper.
> >
> > I believe the chosen method is not sufficient to warrant a vote to accept this paper, but I have increased my score to reflect the author's edits and responses

---

> > > ### Author Response · Authors · 2021-11-23
> > > **Thank you for the discussion**
> > >
> > > Thank you for your comment and for raising the score. We agree that our method is expensive. However, we would like to reiterate that this is the first work on analyzing and annotating latent concepts of a model. In order to encourage and facilitate researchers to work in this direction, we are making the annotation tool and the setup available to the community.
> > >
> > > We would like to clarify that the hierarchical clustering is an informed decision based on the nature of the dataset and the targeted analysis i.e. analyze the learned hierarchy. We experimented with several other algorithms such as K-means, HDBCAN, single linkage and using PCA with hierarchical clustering and found limitations in every setup. E.g. K-means clustering does not identify non-spherical clusters, PCA causes a loss in information and the resulting clusters are not a true reflection of the latent concepts learned by the model.
> > >
> > > We would love to hear about any alternate clustering method that we can use to avoid the mentioned limitation.

---

### Official Review · Reviewer_kVr9 · 2021-11-02

**Correctness:** 3
**Technical Novelty And Significance:** 3
**Empirical Novelty And Significance:** 3
**Recommendation:** 8
**Confidence:** 4

**Main Review:**

Generally, although the method is simple, I think it is effective and solid. Also, I appreciate the efforts in the annotation. I think a direction for improvements is a more in-depth analysis since this paper spends a lot of writing on development (which is fair) but less on analysis. I would suggest simplifying the development details (Section 4.2, 4.4, 6.2 can be shortened, and algorithm 2 can be replaced by a simple textual description) to make space for analysis, specifically:

## Alignments and Differences between discovered and pre-defined concepts

This is a point worth pursuing because the concepts learned in an unsupervised way respect patterns in data much more than human prior definition. I was expecting a detailed analysis about the concepts that are NOT aligned with pre-defined concepts in section 6.2, but there are not and the existing analysis simply reinforces what we have already known. What are these non-aligned concepts? Are they interpretable? Are they more about syntax or semantics? Why they do not align with existing annotation? I strongly encourage the authors to discuss these questions in detail because these are the questions that lead to new insights, rather than reinforcing what we have already familiar with.

## Why latent concepts emerge within BERT

This is another interesting question that may be worth pursuing. This is essentially a question about how large-scale masked language modeling automatically leads to clusters of representations. My intuition is that this should trace back to the effect of contextualization, i.e., clusters emerge because words within the same cluster share the same context, and different clusters originate from different contexts. There could also be other explanations, and I would like to encourage the authors to give a more detailed mechanism about how latent concepts emerge. Also, see [1] as analysis from a manifold perspective.

## Minor comments

- Clustering time and memory consumption should be reported in Section 4.2
- Word examples for each cluster should be provided in the appendix (Table 10)
- Percentages should be provided in table 1

## References

[1] Emergence of Separable Manifolds in Deep Language Representations. Jonathan Mamou, Hang Le, Miguel A Del Rio, Cory Stephenson, Hanlin Tang, Yoon Kim, SueYeon Chung. ICML 2020

**Summary Of The Paper:**

This paper studies the latent concept learned in BERT representations. The authors use a simple and effective clustering method to discover latent concepts in an unsupervised way. The contribution is primarily on the empirical side where the learned concepts are labeled and compared with pre-defined concepts. Then a layer-wise comparison is conducted where the results are similar to previous work but is done in an unsupervised way. The authors summarise this paper by organizing the discovered concepts into a BCD dataset. The writing is also clear. While there are many aspects deserving further pursuit, I think this is a solid paper that makes concrete contributions to the understanding of BERT.

**Summary Of The Review:**

I think this is a solid and interesting paper. My major problem is that this paper spends a lot of writing on development details that could be simplified to make place for more in-depth analysis. I am willing to further increase my score if the authors can present a more in-depth analysis.

---

> ### Author Response · Authors · 2021-11-22
> **Thank you. We have answered the comments and have improved the paper accordingly.**
>
> ***Q: In-depth analysis of clusters not aligned with pre-defined concepts … What are these non-aligned concepts? Are they interpretable? Are they more about syntax or semantics? Why they do not align with existing annotation?***
>
> A: Thank you for the suggestion. We have added a section on unaligned concepts in the paper (See Section 6.4). We found three types of unaligned concepts: i) concepts that are compositional in nature, consisting of more than one fine-grained categories such as VBP and VBD, ii) concepts that cannot be explained via pre-defined categories but are meaningful and interpretable, iii) concepts that are uninterpretable (see Section 6.4 for details and examples of each).
>
> ***Q: … how large-scale masked language modeling automatically leads to clusters of representations …***
>
> A: This is  an interesting question to ponder on. Based on the layer-wise trend seen in Figure 5 and the discussion in section 6.3, we can argue about the emergence of linguistic concepts across layer depth. The model groups words based on similar meaning in the embedding layer and initial layers, and with the presence of context, it forms groupings based on core-linguistic concepts such as POS, Chunking, etc. The meaning-based groups are largely context independent groups while the groups representing core-linguistic concepts are contextual in nature, thus they evolved in the higher layers. We have added discussion on these lines in Section 6.3. Thank you for sharing a very relevant paper. We have discussed it in the paper.
>
> ***Q: Clustering time and memory consumption should be reported in Section 4.2***
>
> A: We have added the information in Section 4.2. The off-the-shelf implementation of Hierarchical clustering has space complexity of O(n^2). For 200,000 points, this implies a matrix of 200,000 * 200,000 elements, which roughly amounts to 320 GB with double precision. The scikit implementation also uses memory for other things internally (maintaining the hierarchical dendrogram etc), and around 400GB is what we have seen experimentally on the dataset used in the paper.
>
> ***Q: Word examples for each cluster should be provided in the appendix (Table 10)***
>
> A: We added Table 11 in the appendix with a few examples of concept labels with words to highlight concept labels and examples of tokens that concept labels represent.
>
> ***Q: Percentages should be provided in table 1***
>
> A: We included percentages in Table 1.

---

> > ### Comment · Reviewer_kVr9 · 2021-11-22
> > **Thank you for your response**
> >
> > I have raised by score.

---

> > > ### Author Response · Authors · 2021-11-23
> > > **Thank you**
> > >
> > > Thank you for acknowledging our response and for raising the score.

---

### Official Review · Reviewer_3Uym · 2021-11-02

**Correctness:** 3
**Technical Novelty And Significance:** 2
**Empirical Novelty And Significance:** 2
**Recommendation:** 5
**Confidence:** 2

**Main Review:**

### Strengths

The problem that the paper tackles, analysing latent concepts captured in BERT, is challenging and needed to understand the model. It is well in line with the trend of model and representation interpretation.

The proposed solution to the problem requires carefully manually annotation which was carried out thoughtfully. The innovation is that there are several syntactic and semantics aspects taken into account, including hierarchy nature of those concepts.

The analyses presented in the paper do lead to several findings about the knowledge of BERT, thus they back the claims risen in the paper.


### Weaknesses

The paper doesn't go any further beyond 1000 clusters / concepts, which doesn't seem to have as wide coverage as e.g. wikipedia. Hence, one would question about the used clustering method, especially whether it is good enough to discover a wide range of concepts / clusters.

The methodology proposed in the paper requires expensive manual annotation. This leads to the problem of replication. Besides, although the BCD dataset has some potential, it is limited to only BERT. It does't seem useful when other models are examined.

I found the BCD development (section 7.1) problematic. Although clusters were manually annotated, it is unclear how accurate an instance is assigned to a specific cluster. For instance, although annotators intuitively agreed that cluster A with N instances properly present concept "animal", how many instances among the N instances actually belong to concept "animal"?






**Summary Of The Paper:**

The paper analyses concepts that BERT can capture. The core idea is to use agglomerative hierarchical clustering method to discover latent concepts (i.e. clusters) with hierarchy. Those clusters were manually assigned labels / meanings. The authors then used these labelled clusters to analyse concepts that BERT captures, across several syntactic and semantics aspects. The labelled clusters were also used to build a concept dataset (namely BERT concept dataset - BCD) for 1M instances (i.e. words in contexts).

**Summary Of The Review:**

The paper tackles an important challenge with thoughtful methodology and reasonable analyses. The paper is also well written, presenting several findings about the knowledge captured in BERT.
The proposed methodology however has problems which limit the contribution of the paper.

---

> ### Author Response · Authors · 2021-11-22
> **Thank You. We have answered the comments and have improved the paper accordingly.**
>
> ***Q: The paper doesn't go any further beyond 1000 clusters…***
>
> A: We experimented with a range of clusters varying between 200-1600 and empirically selected 1000 clusters using Elbow.  We found that using more than 1000 clusters resulted in over-clustering, with a large number of small clusters consisting of less than 5 word types. On the other hand, relatively fewer clusters resulted in very large clusters that are hard to interpret. Using 1000 clusters provides a good balance between under-clustering and over-clustering.
>
> ***Q: The methodology proposed in the paper requires expensive manual annotation. This leads to the problem of replication.***
>
> A: We agree that the annotation effort is expensive. However, without such regressive annotation it is difficult to accurately analyze the latent concepts learned in pre-trained models. To reduce the future annotation efforts, we developed an annotation platform which provides a visualization of clusters, along with context of each word and automatic matching to predefined labels and the BCD labels. Moreover, in order to extend the annotation to other models, we can automatically map BCD labels to other models, say RoBERTa, using a distant supervision approach.
>
> ***Q: Besides, although the BCD dataset has some potential, it is limited to only BERT. It does't seem useful when other models are examined.***
>
> A: We chose BERT since it is one of the most widely used models. We agree that the annotations were carried out for BERT only. But as we have mentioned above, annotation of any new models can greatly benefit from the annotations carried in this work. Moreover, BCD can be used as a supervised set (similar to POS and SEM tagging) to interpret any pretrained model against fine-grained categories, similar to the use case shown in the paper.
>
> ***Q: I found the BCD development (section 7.1) problematic… how accurate an instance is assigned to a specific …***
>
> A: The annotators would not mark a cluster as a concept, say “animal”, unless the majority  (> 90%) of the words are referring to animals. Note that annotators see all words in the cluster along with their context while annotating. Each annotation is carried out by 3 linguists and 3 consolidators. The inter-annotator agreement is “substantial” in Kappa measurement. Based on your suggestion, we manually verified the 10% annotated concepts. We found that on average 92% of the words indeed belonged to the assigned concept.

---

> > ### Comment · Reviewer_3Uym · 2021-11-23
> > **thanks for the response**
> >
> > Thanks the authors for the informative response.
> >
> > I would like to clarify my criticisms that I would like to hear more from the authors.
> >
> > The first problem is the number of clusters. I do understand the technical point of choosing 1000. I'm however not convinced that the number was chosen reasonably. In Fig 2 we can see a very sudden drop from 950 to 1000. I don't believe that adding 50 more clusters could yield such a big change. This means that the judgement of selecting 1000 is due to the abnormal behaviour of the used clustering algorithm rather than any scientific logics. Also, as I mentioned in the review, 1000 concepts hardly have a proper wide coverage. (This is especially true if we think about the number of word senses - each of which can be considered as a semantic concept.)
> >
> > The second problem is annotation cost and effort. I don't think the annotation approach in the paper could give a good picture. Even with only 1000 clusters, the annotation covered only 23%. In other words, 77% are missed. What happens if we have 10x more clusters? This indicates that approach is very poorly scalable.
> >
> > The two points lead to the question about the usability of BCD: is its coverage wide enough given only 198 / 1000 concepts are labeled? Now, if one uses RoBERT instead of BERT and wants to use BCD (with distant learning as the authors suggested), only 19% (with a very rough estimation) of words can be mapped.

---

> > > ### Author Response · Authors · 2021-11-24
> > > **thanks for the discussion**
> > >
> > > Thank you for your response.
> > >
> > > These are valid arguments. Our decision to select 1000 clusters was not only based on empirical evidence (Figure 2) but also qualitative observations made on various number of clusters. We agree that 1000 clusters may not be the optimal to cover all possible senses. However, given the hierarchical nature of language, a 1000 categories may already show a good coarse-level picture of the concepts learned by BERT. Our study is first of its kind, and we hope to extend it to more fine-grained concepts by applying multi-level clustering or combining findings from multiple clustering runs on different subsets of the data.
> > >
> > > On the usability of BCD: We agree that manual annotation of the concepts is costly -- but we believe that it is important to annotate the concepts learned within these models to accurately interpret them. Now that we have streamlined the framework, we hope to scale the effort using crowdsourcing. Additionally, as the number of annotations increases, distant learning for other models will become easier.

---

### Decision · Program_Chairs · 2022-01-20

**Decision:**

Accept (Poster)

**Comment:**

This paper analyzes the latent concepts learned in BERT. In contrast to previous work which tries to map embeddings to predefined
linguistic concepts this paper sets out to discover what is inherently learned by BERT. This is however easier said that done, since
there is no easy way to inspect the embeddings and draw conclusions on what is being learned. The authors adopt a methodology which could be used to inspect the inner workings of other pretrained models. They employ hierarchical clustering to discover latent concepts and then inspect these clusters by manually labeling them. The reviewers raised various issues regarding the number of clusters, and the amount of effort required which de facto renders the approach not very portable. The authors addressed the comments and flagged several difficulties with undertaking such an analysis. I will vote for the paper to be presented as an oral for two reasons a) it is difficult to analyze pretrained models, and although I am not convinced what the authors propose is feasible, it will at least get the discussion going, b) the manually annotated dataset is useful and will go towards allowing us to perform comparisons between models c) the annotation tool will be useful to others if the authors are considering releasing it.